# Path-Decoupled Hyperbolic Flow Matching for Few-Shot Adaptation

**Lin Li** [1,2]  **Ziqi Jiang** [2]  **Gefan Ye** [3]  **Zhenqi He** [2]  **Jiahui Li** [3]  **Jun Xiao** [3]  **Kwang-Ting Cheng** [1,2]  **Long Chen** [2]

## Abstract

Recent advances in cross-modal few-shot adaptation treat visual-semantic alignment as a continuous feature transport problem via Flow Matching (FM). However, we argue that Euclidean-based FM overlooks fundamental limitations of flat geometry, where polynomial volume growth fails to accommodate diverse feature distributions, leading to severe **path entanglement**. To this end, we propose path-decoupled Hyperbolic Flow Matching (HFM), leveraging the Lorentz manifold's exponential expansion for trajectory decoupling. HFM structures the transport via two key designs: 1) *Centripetal hyperbolic alignment*: It constructs a centripetal hierarchy by anchoring textual roots, which pushes visual leaves to the boundary to initialize orderly flows. 2) *Path-decoupled objective*: It acts as a "semantic guardrail" rigidly confining trajectories within isolated class-specific geodesic corridors via step-wise supervision. Furthermore, we devise an adaptive *diameter-based stopping* to prevent over-transportation into the crowded origin based on the intrinsic semantic scale. Extensive ablations on 11 benchmarks have shown that HFM establishes a new state-of-the-art, consistently outperforming its Euclidean counterparts. Codes are available at here.

## 1. Introduction

The remarkable zero-shot generalization of pretrained vision-language models (VLMs), such as CLIP (Radford et al., 2021), builds upon their ability to embed images and natural language into a shared semantic embedding space (Jia et al., 2021; Li et al., 2022; Zhong et al., 2022). Although such joint representations facilitate zero-shot classification via cross-modal similarity, a large performance

[1]AI Chip Center for Emerging Smart Systems, Hong Kong SAR, China [2]The Hong Kong University of Science and Technology (HKUST) [3]Zhejiang University. Correspondence to: Long Chen <longchen@ust.hk>.

*Proceedings of the 43$^{rd}$ International Conference on Machine Learning*, Seoul, South Korea. PMLR 306, 2026. Copyright 2026 by the author(s).

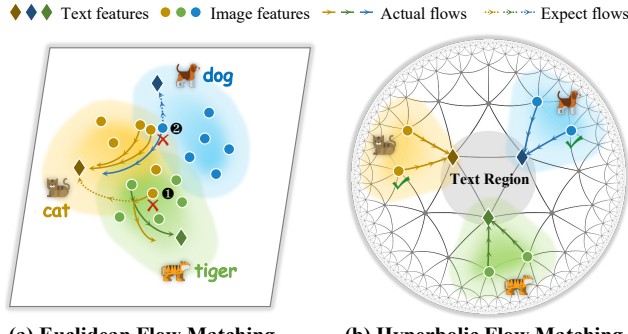

*Figure 1.* Illustration of Path Entanglement. (a) Euclidean Flow Matching suffers from severe trajectory collisions (*e.g.*, "cat" intersecting "tiger" and "dog" merging with "cat") due to the limited polynomial capacity of flat geometry. (b) Hyperbolic Flow Matching uses exponential volume expansion to achieve path decoupling.

gap remains when encountering specialized downstream tasks (Zhou et al., 2022b; Zhang et al., 2022). To bridge this gap, few-shot adaptation (Zhou et al., 2022b;a; Zhang et al., 2022; Gao et al., 2024) aims to refine the pre-trained encoders, leveraging minimal supervisory signals to realign visual features with their corresponding textual prototypes.

To achieve such adaptation, various Parameter-Efficient Fine-Tuning (PEFT) techniques, *e.g.*, prompt tuning (Zhou et al., 2022b;a) and feature-level adapters (Gao et al., 2024; Hu et al., 2022a; Zanella & Ben Ayed, 2024), have been widely adopted to refine latent representations. However, these methods predominantly rely on a "one-step" adjustment, where the alignment is restricted to a single forward pass of the adapter module. Such a direct transformation often struggles to resolve complex semantic entanglements in challenging datasets, due to the lack of the iterative rectification inherent in continuous processes (Jiang et al., 2025). To address this, recent efforts like FMA (Jiang et al., 2025) treat alignment as a continuous feature transport problem via Flow Matching (FM) (Lipman et al., 2022; Liu et al., 2022; He et al., 2026) strategy. Specifically, they construct conditional flow paths that guide visual features from the image manifold to the corresponding class-specific text regions, thereby eliminating the cross-modal gap for classification. By learning a velocity field that defines a time-continuous evolution, the aforementioned methods enable multi-step rectification. This iterative refinement allows the model to progressively correct alignment errors, enabling superior

expressive capacity over one-step counterparts.

Despite the theoretical flexibility of multi-step flows, we argue that existing Euclidean FM models suffer from severe inherent **path entanglement**. It refers to the transport paths that inadvertently intersect, overlap, or merge within the latent space. Fundamentally, this entanglement arises because the polynomial volume growth of flat geometry fails to accommodate diverse feature distributions, leading to two inevitable structural interferences: ❶ *Disordered Cross-Modality Flows*. Source image features and their corresponding target semantic prototype are embedded in *irregularly dispersed* positions. Bridging this unstructured gap necessitates long-range transport, thereby increasing the risk of trajectory collisions (*e.g.*, the yellow flow of the "cat" colliding with and following the green path of the "tiger" in Figure 1a). This chaotic confluence erodes feature discriminability, thereby compromising the classification performance. ❷ *Crowded Inter-Class Flows*. Flow sources from different categories may unintentionally overlap, leading to more ambiguous transport trajectories. Without sufficient separation, these paths are easily diverted by *high-density neighboring clusters* (*e.g.*, the blue flow of "dog" drifting into the dense yellow samples in Figure 1a).

To this end, in this paper, we propose a new path-decoupled **H**yperbolic **F**low **M**atching (**HFM**) for few-shot adaptation. By reformulating the transport dynamics within the Lorentz manifold, we exploit the exponential volume growth of hyperbolic geometry to spatially decouple the transport trajectories to alleviate path entanglement. To address ❶, we employ a mechanism of **centripetal hyperbolic alignment**. It restructures the latent geometry into a concentric hierarchy by explicitly initializing textual prototypes with small hyperbolic radii (proximal to the origin, visualized as the central "Text Region" in Figure 1b) and visual features near the manifold boundary. To enforce this *centripetal order*, we leverage the *cross-modality entailment* objective (Desai et al., 2023) to optimize the embedding space, formally establishing text as the semantic root and images as entailment leaves. By constraining centripetal order flows to focusing radial geodesics, we can reduce trajectory collisions, thereby preserving semantic discriminability and enhancing classification performance. To tackle ❷, we introduce a **path-decoupled objective**. Leveraging the exponential volume expansion near the manifold boundary, this objective maximizes the geodesic margin between distinct semantic categories. This optimization explicitly constructs isolated geodesic transport corridors (illustrated as the separated yellow, green, and blue areas in Figure 1b), ensuring that each class-specific flow evolves within a unique, non-overlapping volume. Consequently, this geometric separation effectively decouples inter-class trajectories, preventing the ambiguous overlaps inherent to crowded Euclidean spaces.

Furthermore, in the inference stage, to enhance inference efficiency and accuracy, we devise an adaptive **diameter-based stopping** strategy. By dynamically terminating the flow upon the convergence of geometric compactness, we prevent over-transportation into the densely populated text region near the origin. This avoids the critical risk of visual features drifting into adjacent, incorrect clusters due to spatial crowding, ensuring both precise classification and minimal computational overhead.

Extensive experiments on 11 few-shot benchmarks demonstrate that our HFM consistently surpasses state-of-the-art Euclidean-based FM counterparts and PEFT adapters as a plug-and-play module. These promising results evidence the significant potential of non-Euclidean manifolds for advancing the frontier of few-shot cross-modal understanding.

## 2. Related Works

**Few-shot Adaptions in VLMs.** Few-shot adaptation is a task that requires models to learn from only a few annotated examples (Chen et al., 2020; Hou et al., 2019; Wang et al., 2020). Current approaches in VLMs usually follow the paradigm that employs pre-trained VLMs and then refines them through task-adaptive optimization. Parameter-Efficient Fine-Tuning (PEFT) (Gu et al., 2022; Hu et al., 2023; Lee et al., 2023; Zhang et al., 2022; Zanella & Ben Ayed, 2024; De Marinis et al., 2025) is one of popular optimization methods, which focuses on updating a small subset of parameters to achieve performance comparable to full fine-tuning. In the meanwhile, some studies explore the potential of in-context learning (Alayrac et al., 2022; Zhang et al., 2023; Hu et al., 2022b; Cai et al., 2023; Huang et al., 2024) in few-shot adaptation. With no change on model parameters, they add support examples into the inference context to make pre-trained models adapt to specific tasks. Despite computational efficiency and effectiveness, their one-step adjustment fails to disentangle complex feature distributions, leaving substantial room for improvement.

**Hyperbolic Representation Learning.** Hyperbolic geometry, such as Poincaré-ball Model (Nickel & Kiela, 2017) and Lorentz Model (Nickel & Kiela, 2018), demonstrates extraordinary capability in modeling hierarchical relationships due to its exponential volume growth relative to radius. With custom-designed neural network modules (Chen et al., 2022b; Liu et al., 2019; Li et al., 2025c), hyperbolic representation learning has achieved profound breakthroughs in a wide range of tasks in different modalities, including text (Dhingra et al., 2018; Zhu et al., 2020; Le et al., 2019), image (Atigh et al., 2022; Khrulkov et al., 2020; Li et al., 2023), audio (Petermann et al., 2023; Nakashima et al., 2022) and video (Li et al., 2025b;a). Meanwhile, combining with contrastive learning methods enables hyperbolic representation learning to successfully adapt to cross-modal

tasks (Wen et al., 2025; Desai et al., 2023; Liu et al., 2020; Hong et al., 2023). Given the inherent cross-modality hierarchy of the data in few-shot adaptation task, we try to harness the power of hyperbolic geometry for flow matching.

**Flow Matching (FM).** Diffusion Model (Ho et al., 2020; Song et al., 2020; Rombach et al., 2022) is a powerful generative tool, but limited by slow inference due to curved sampling paths. Flow Matching (Lipman et al., 2022; Liu et al., 2022) addresses this limitation by learning vector fields that generate straight-line trajectories between distributions. Specifically, the utilization of optimal transport simplifies the geometric structure of the vector field, thus facilitating faster and more accurate sampling. From the perspective of stochastic interpolants, FM and diffusion models act as special cases within a unifying framework (Albergo et al., 2023). In spite of remarkable successes in domains like image generation (Ren et al., 2024; Esser et al., 2024; Luo et al., 2025), FM has recently been introduced to cross-modal adaptation. Notable works like FMA (Jiang et al., 2025) treat visual-semantic alignment as a continuous feature transport problem. However, constrained by flat geometry, these Euclidean-based FM often suffer from path entanglement, impairing the classification performance.

## 3. Approach

In this section, we present HFM, a path-decoupled framework designed to alleviate the path entanglement in few-shot adaptation. As illustrated in Figure 2, our HFM comprises three phases: 1) **Constructing Centripetal Hyperbolic Space** (§3.2): Establishing a centripetal hierarchy by anchoring textual roots at the origin and visual leaves at the boundary. 2) **Learning Path-Decoupled Flows** (§3.3): Optimizing a tangent velocity field to confine transport within isolated geodesic corridors via step-wise and contrastive supervision. 3) **Inference with Diameter-based Stopping** (§3.4): Employing a density-aware termination criterion to prevent over-transportation in crowded text region.

### 3.1. Preliminaries: Lorentz Model

We first briefly introduce the key concepts of hyperbolic geometry on which our HFM relies. Following (Desai et al., 2023; Pal et al., 2025), we adopt the Lorentz model of hyperbolic space $\mathbb{L}^{n,\kappa}$ with *learnable* constant negative curvature $-\kappa$ ($\kappa > 0$). In this model, each point $\boldsymbol{x} \in \mathbb{L}^{n,\kappa}$ consists of a time-like scalar $x_0$ and a space-like vector $\tilde{\boldsymbol{x}}$, constrained by the geometric property $\langle \boldsymbol{x}, \boldsymbol{x} \rangle_{\mathbb{L}} = -1/\kappa$. The Lorentzian inner product is formulated as $\langle \boldsymbol{x}, \boldsymbol{y} \rangle_{\mathbb{L}} = -x_0 y_0 + \langle \tilde{\boldsymbol{x}}, \tilde{\boldsymbol{y}} \rangle_{\mathbb{E}}$, where $\langle \cdot, \cdot \rangle_{\mathbb{E}}$ is the Euclidean inner product.

**Geodesics.** The shortest path between two points on the manifold, written as $d_{\mathbb{L}}(\boldsymbol{x}, \boldsymbol{y}) = \frac{1}{\sqrt{\kappa}} \text{arcosh}(-\kappa \langle \boldsymbol{x}, \boldsymbol{y} \rangle_{\mathbb{L}})$.

**Tangent and Manifold Projection.** To map Euclidean

features from CLIP into $\mathbb{L}^{n,\kappa}$, we employ the *exponential map* (Nickel & Kiela, 2018) defined at the tangent space of a reference point $\boldsymbol{z}$. To align with this geometry, the original Euclidean feature $\tilde{\boldsymbol{x}}_{\mathbb{E}} \in \mathbb{R}^n$ is lifted into the $(n+1)$-dimensional ambient tangent space $T_{\boldsymbol{z}}\mathbb{L}$, satisfying the orthogonality constraint with $\boldsymbol{z}$. Consequently, the projection onto the manifold to obtain $\boldsymbol{x} \in \mathbb{L}^{n,\kappa}$ is formulated as:

$$\boldsymbol{x} = \exp_{\boldsymbol{z}}^{\kappa}(\boldsymbol{x}_{\mathbb{E}}) = \cosh(\sqrt{\kappa}\|\boldsymbol{x}_{\mathbb{E}}\|_{\mathbb{L}})\boldsymbol{z} + \frac{\sinh(\sqrt{\kappa}\|\boldsymbol{x}_{\mathbb{E}}\|_{\mathbb{L}})}{\sqrt{\kappa}\|\boldsymbol{x}_{\mathbb{E}}\|_{\mathbb{L}}}\boldsymbol{x}_{\mathbb{E}}, \quad (1)$$

where $\boldsymbol{x}_{\mathbb{E}}$ refers to the lifted ambient vector derived from $\tilde{\boldsymbol{x}}_{\mathbb{E}}$. Conversely, the *logarithmic map* serves as the inverse operation, projecting a point $\boldsymbol{x}$ from the manifold back to the tangent space at $\boldsymbol{z}$ to recover the Euclidean features:

$$\boldsymbol{x}_{\mathbb{E}} = \log_{\boldsymbol{z}}^{\kappa}(\boldsymbol{x}) = \frac{\text{arccosh}(-\kappa \langle \boldsymbol{z}, \boldsymbol{x} \rangle_{\mathbb{L}})}{\sqrt{\langle \boldsymbol{z}, \boldsymbol{x} \rangle_{\mathbb{L}}^2 - 1/\kappa^2}} \Pi_{T_{\boldsymbol{z}}\mathbb{L}}(\boldsymbol{x}), \quad (2)$$

where $\Pi_{T_{\boldsymbol{z}}\mathbb{L}}(\boldsymbol{x}) = \boldsymbol{x} + \kappa \langle \boldsymbol{z}, \boldsymbol{x} \rangle_{\mathbb{L}} \boldsymbol{z}$ denotes the orthogonal projection onto the tangent space.

### 3.2. Constructing Centripetal Hyperbolic Space

To address disordered cross-modality flows, we first align the latent geometry into a *centripetal hierarchy* (Figure 2a). Formally, let $\boldsymbol{x}_1$ denote the hyperbolic textual prototypes (serving as semantic roots/targets) and $\boldsymbol{x}_0$ denote the hyperbolic visual features (serving as entailment leaves/sources). This alignment resolves path entanglement by transforming arbitrary transport paths into ordered centripetal flows from $\boldsymbol{x}_0$ to $\boldsymbol{x}_1$. The exponential boundary expansion guarantees sufficient initialization spacing for visual feature $\boldsymbol{x}_0$, reducing spatial trajectory overlap during inward transport.

**Geometric Stratification.** We explicitly impose a centripetal hierarchy by initializing learnable scalars as: $\alpha_{\text{txt}} < \alpha_{\text{img}}$ to modulate the feature norm before tangent to manifold projection. This configuration anchors text near the origin and pushes images to the boundary (§Figure 2a), creating an explicit geometric prior for inward transport.

**Centripetal hyperbolic alignment.** To solidify this hierarchy, we optimize the embeddings using joint objectives that enforce both geometric entailment and semantic discrimination. In Figure 2a, we employ the *hyperbolic entailment loss* (Desai et al., 2023) to enforce a partial order where the text prototype $\boldsymbol{x}_1$ (parent) spatially entails the image feature $\boldsymbol{x}_0$ (child). Formally, we constrain $\boldsymbol{x}_0$ to lie within the entailment cone of $\boldsymbol{x}_1$ by minimizing the angular violation:

$$\mathcal{L}_{\text{entail}} = \max\left(0, \pi - \angle \boldsymbol{0}\boldsymbol{x}_1\boldsymbol{x}_0 - \omega(\boldsymbol{x}_1)\right), \quad (3)$$

where the term $\pi - \angle \boldsymbol{0}\boldsymbol{x}_1\boldsymbol{x}_0$ corresponds to the exterior angle at $\boldsymbol{x}_1$, explicitly quantifying the deviation of the child $\boldsymbol{x}_0$ from the parent's radial axis. The term $\omega(\boldsymbol{x}_1) = \arcsin(2H/\|\boldsymbol{x}_1\|_{\mathbb{L}})$ represents the cone aperture, which naturally narrows as the prototype's radius increases.

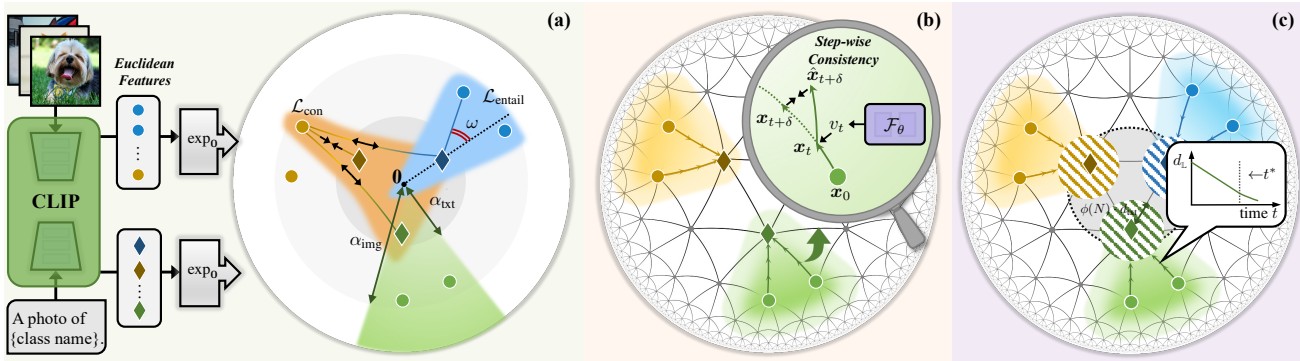

*Figure 2.* The overview of HFM. (a) **Constructing Centripetal Hyperbolic Space**: Establish a centripetal cross-modal hierarchy, optimizing textual roots near the origin and visual leaves toward the boundary. (b) **Learning Path-Decoupled Flows**: Tangent velocity fields $\mathcal{F}_\theta$ are optimized to guide features along isolated geodesic corridors. The step-wise consistency enforces trajectory decoupling in hyperbolic space. (c) **Inference with Diameter-based Stopping**: The flow terminates at $t^*$ once the distance to text prototypes drops below a dynamic threshold scaled by semantic diameter $d_{\text{txt}}$ to alleviate over-transportation into the crowded origin.

Complementing this, we use a hyperbolic contrastive loss $\mathcal{L}_{\text{con}}$ to enforce semantic discrimination. It minimizes the geodesic distance between the image feature $\boldsymbol{x}_0$ and its ground-truth text prototype $\boldsymbol{x}_1^c$, while maximizing distances to other prototypes in the support set:

$$\mathcal{L}_{\text{con}} = -\log \frac{\exp\left(-d_{\mathbb{L}}(\boldsymbol{x}_0, \boldsymbol{x}_1^c)/\tau\right)}{\sum_k \exp\left(-d_{\mathbb{L}}(\boldsymbol{x}_0, \boldsymbol{x}_1^k)/\tau\right)}. \quad (4)$$

### 3.3. Learning Path-Decoupled Flows

Having aligned the centripetal hierarchy, we proceed to learn the transport dynamics that guide entailment from visual leaves to semantic roots. Unlike standard Riemannian flow matching (Chen & Lipman, 2022) that typically requires integrating an ordinary differential equation (ODE) over a continuous velocity field, we propose path-decoupled HFM. Our core insight is to exploit *step-wise transport* (Yang et al., 2024) to enable explicit geometric supervision (§Figure 2b).

**Geodesic Path.** Similar to the plain Euclidean flow matching (Jiang et al., 2025), we define the ground-truth trajectory as the geodesic path on $\mathbb{L}$ connecting the source image $\boldsymbol{x}_0$ explicitly to its ground-truth class prototype $\boldsymbol{x}_1$. This strict pairing disentangles the optimization into independent source-target tasks, effectively eliminating inter-class interference (Jiang et al., 2025).

**Tangent Velocity Alignment.** To learn the transport dynamics, we sample paired states $(\boldsymbol{x}_t, \boldsymbol{x}_{t+\delta})$ from the ground-truth geodesic path. Specifically, given a random time $t$ and a step size $\delta$, we aim to predict the update that transports the feature from $\boldsymbol{x}_t$ to $\boldsymbol{x}_{t+\delta}$. Our network $\mathcal{F}_\theta$ predicts an ambient vector at state $\boldsymbol{x}_t$, which is explicitly projected onto the local tangent space to ensure geometric validity:

$$\boldsymbol{v}_t = \Pi_{T_{\boldsymbol{x}_t}\mathbb{L}}\left(\mathcal{F}_\theta(\boldsymbol{x}_t, t)\right), \quad (5)$$

where $\Pi_{T_{\boldsymbol{x}_t}\mathbb{L}}$ denotes the orthogonal projection onto the tangent space at $\boldsymbol{x}_t$. We then model the discrete evolution

by mapping this tangent velocity back onto the manifold via the exponential map (Chen & Lipman, 2022):

$$\hat{\boldsymbol{x}}_{t+\delta} = \exp_{\boldsymbol{x}_t}^{\kappa}(\delta \cdot \boldsymbol{v}_t). \quad (6)$$

This parameterization ensures that the predicted next state $\hat{\boldsymbol{x}}_{t+\delta}$ strictly resides on the hyperbolic manifold $\mathbb{L}$.

**Path-decoupled Objective.** We optimize the network using a path-decoupled objective that enforces both step-wise trajectory consistency and semantic separation. Crucially, both losses operate on the predicted future state $\hat{\boldsymbol{x}}_{t+\delta}$, enabling direct rectification of the flow.

*1) Step-wise Consistency Loss:* To ensure the learned flow precisely follows the geodesic path, we minimize the squared Riemannian distance between the predicted state $\hat{\boldsymbol{x}}_{t+\delta}$ and the ground-truth target $\boldsymbol{x}_{t+\delta}$:

$$\mathcal{L}_{\text{step}} = \|d_{\mathbb{L}}(\hat{\boldsymbol{x}}_{t+\delta}, \boldsymbol{x}_{t+\delta})\|^2. \quad (7)$$

*2) Inter-Class Decoupling Loss:* To prevent the flow from drifting into the attraction basins of incorrect classes, we impose a "semantic guardrail" via a dynamic hyperbolic contrastive loss. Unlike the static alignment, this objective forces the predicted intermediate state $\hat{\boldsymbol{x}}_{t+\delta}$ to maximize its similarity with the corresponding prototype $\boldsymbol{x}_1^c$ while repelling all negative prototypes:

$$\mathcal{L}_{\text{icd}} = -\log \frac{\exp\left(-d_{\mathbb{L}}(\hat{\boldsymbol{x}}_{t+\delta}, \boldsymbol{x}_1^c)/\tau\right)}{\sum_k \exp\left(-d_{\mathbb{L}}(\hat{\boldsymbol{x}}_{t+\delta}, \boldsymbol{x}_1^k)/\tau\right)}. \quad (8)$$

The total objective is $\mathcal{L} = \mathcal{L}_{\text{step}} + \lambda\mathcal{L}_{\text{icd}}$. The overall training procedure is summarized in Algorithm 1. By penalizing semantic deviations at each intermediate step, this mechanism strictly confines trajectories to class-specific corridors, thereby eliminating the path entanglement issue.

### 3.4. Inference with Diameter-based Stopping

During inference, we transport the visual features of test images along the predicted geodesic flow. To prevent over-

---

**Algorithm 1** Training Path-Decoupled Flows

---

1: **Input:** paired hyperbolic image features and text features $\mathcal{D} = \{(\boldsymbol{x}_0, \boldsymbol{x}_1)\}$, stepsize $\delta$, weight $\lambda$.
2: **Initialize:** flow network parameters $\theta$.
3: **repeat**
4:      $t \sim \mathcal{U}([0,1]), (\boldsymbol{x}_0, \boldsymbol{x}_1) \sim \mathcal{D}$
5:      $\boldsymbol{x}_t \leftarrow \exp_{\boldsymbol{x}_0}^{\kappa}(t \cdot \log_{\boldsymbol{x}_0}^{\kappa}(\boldsymbol{x}_1))$
6:      $\boldsymbol{x}_{t+\delta} \leftarrow \exp_{\boldsymbol{x}_0}^{\kappa}((t + \delta) \cdot \log_{\boldsymbol{x}_0}^{\kappa}(\boldsymbol{x}_1))$
7:      $\boldsymbol{v}_t \leftarrow \Pi_{T_{\boldsymbol{x}_t}\mathbb{L}}(\mathcal{F}_{\theta}(\boldsymbol{x}_t, t))$
8:      $\hat{\boldsymbol{x}}_{t+\delta} \leftarrow \exp_{\boldsymbol{x}_t}^{\kappa}(\delta \cdot \boldsymbol{v}_t)$
9:      $\mathcal{L} = \|d_{\mathbb{L}}(\hat{\boldsymbol{x}}_{t+\delta}, \boldsymbol{x}_{t+\delta})\|^2 + \lambda\mathcal{L}_{\text{icd}}$
10:     Update $\theta$ using gradient descent to minimize $\mathcal{L}$
11: **until** converged

---

**Algorithm 2** Inference with Diameter-based Stopping

---

1: **Input:** image feature $\boldsymbol{x}_0$, text prototypes $\{\boldsymbol{x}_1^c\}$, trained flow network $\mathcal{F}_{\theta}$, stepsize $\delta$, class number $N$.
2: **Initialize:** $t \leftarrow 0$, $d_{\text{txt}} = \max_{i,j} d_{\mathbb{L}}(\boldsymbol{x}_1^i, \boldsymbol{x}_1^j)$.
3: **while** $t < 1$ **do**
4:      $\boldsymbol{v}_t \leftarrow \Pi_{T_{\hat{\boldsymbol{x}}_t}\mathbb{L}}(\mathcal{F}_{\theta}(\hat{\boldsymbol{x}}_t, t))$
5:      $\hat{\boldsymbol{x}}_{t+\delta} \leftarrow \exp_{\hat{\boldsymbol{x}}_t}^{\kappa}(\delta \cdot \boldsymbol{v}_t)$
6:      **if** $\min_c d_{\mathbb{L}}(\hat{\boldsymbol{x}}_{t+\delta}, \boldsymbol{x}_1^c) \leq \phi(N) \cdot d_{\text{txt}}$ **then**
7:         $t^* \leftarrow t + \delta$; **break**
8:      **end if**
9:      $t \leftarrow t + \delta$
10: **end while**
11: **return** $\hat{y} = \arg\min_c \sum_{t=0}^{t^*} d_{\mathbb{L}}(\hat{\boldsymbol{x}}_t, \boldsymbol{x}_1^c)$

---

shooting at the crowded hyperbolic boundary, we further propose an adaptive strategy comprising two steps:

**Riemannian Euler Integration.** We solve the transport ODE using a discrete Euler method (Lipman et al., 2022). Specifically, starting from the hyperbolic visual feature $\hat{\boldsymbol{x}}_0$, we iteratively update the state by projecting the predicted tangent velocity back onto the manifold:

$$\hat{\boldsymbol{x}}_{t+\delta} = \exp_{\hat{\boldsymbol{x}}_t}^{\kappa}\left(\delta \cdot \Pi_{T_{\hat{\boldsymbol{x}}_t}\mathbb{L}}(\mathcal{F}_{\theta}(\hat{\boldsymbol{x}}_t, t))\right). \quad (9)$$

**Diameter-based Stopping.** We define the semantic diameter $d_{\text{txt}}$ as the maximum pairwise geodesic distance among all target prototypes $\{\boldsymbol{x}_1^c\}$. During inference, transport terminates at step $t^*$ once the geodesic distance to the nearest prototype falls within a density-adaptive threshold:

$$\min_c d_{\mathbb{L}}(\hat{\boldsymbol{x}}_{t^*}, \boldsymbol{x}_1^c) \leq \phi(N) \cdot d_{\text{txt}}. \quad (10)$$

Here, $d_{\text{txt}} = \max_{i,j} d_{\mathbb{L}}(\boldsymbol{x}_1^i, \boldsymbol{x}_1^j)$ represents the intrinsic semantic scale, while $\phi(N) = 0.5 \log_{10}(N)$ compensates for manifold crowding as the class cardinality $N$ grows.

To mitigate local fluctuations, instead of relying solely on the final state, we ensemble the class probabilities across all

valid steps up to $t^*$ to determine the final prediction. The entire inference process is presented in Algorithm 2.

# 4. Experiments

**Datasets.** We evaluated HFM on standard few-shot image classification. Particularly, we conducted experiments on 11 benchmarks, including Aircraft (Maji et al., 2013), EuroSAT (Helber et al., 2019), DTD (Cimpoi et al., 2014), SUN397 (Xiao et al., 2010), UCF101 (Soomro et al., 2012), StanfordCars (Krause et al., 2013), ImageNet (Deng et al., 2009), Flowers102 (Nilsback & Zisserman, 2008), Food101 (Bossard et al., 2014), OxfordPets (Parkhi et al., 2012), and Caltech101 (Fei-Fei et al., 2004). Following prior works (Jiang et al., 2025), we partitioned them into two subsets: the first five datasets form the *difficult* group, while the remaining six constitute the *easy* group. For each dataset, we adopted the standard train/validation/test splits. Under the $K$-shot setting, we constructed the training set by uniformly sampling $K$ labeled images per class, with the rest used for validation and testing according to the protocol.

**Implementation Details.** We implemented HFM on top of CLIP-LoRA (Zanella & Ben Ayed, 2024). Specifically, we first utilized the trained CLIP-LoRA model after centripetal hyperbolic alignment to extract image and text features, then trained a velocity network on these features following the HFM framework. Similar to FMA (Jiang et al., 2025), we adopted the lightweight architecture from MAR (Li et al., 2024) as our flow-matching network, implemented as a deep residual MLP with timestep conditioning. As for the Lorentz model, we initialized the learnable manifold curvature $\kappa = 1.0$. Following (Desai et al., 2023), we set a constant $H{=}0.1$. The balance weight $\lambda$ for the inter-class decoupling loss was set to 0.1. We optimized HFM with AdamW (Loshchilov & Hutter, 2017) using a learning rate of $2 \times 10^{-4}$ and a cosine annealing schedule.

## 4.1. Quantitative Comparison

**Settings.** We compared HFM with recent few-shot classification methods including CLIP (Radford et al., 2021), CoOp (Zhou et al., 2022b), CoCoOp (Zhou et al., 2022a), TIP-Adapter (Zhang et al., 2022), CLIP-Adapter (Gao et al., 2024), PLOT++ (Chen et al., 2022a), KgCoOp (Yao et al., 2023), ProGrad (Zhu et al., 2023), CLIP-LoRA (Zanella & Ben Ayed, 2024) and FMA (Jiang et al., 2025). All methods use CLIP ViT-B/16 (Radford et al., 2021) as the backbone.

**Results.** As seen from Table 1, HFM consistently outperforms SOTA baselines across all settings. Crucially, HFM surpasses the Euclidean counterpart FMA (Jiang et al., 2025), verifying the efficacy of hyperbolic geometry in resolving path entanglement. On the difficult benchmarks, HFM achieves **64.1%** (1-shot) and **79.8%** (16-shot), ex-

*Table 1.* **Quantitative results.** Performance comparison (§4.1) of the state-of-the-art approaches. Based on CLIP-LoRA, we add HFM to further improve the performance. Top-1 accuracy averaged over 3 random seeds is reported. The highest value of each dataset is bolded.

| Shots | Method | Difficult | | | | | | Easy | | | | | | |
|---|---|---|---|---|---|---|---|---|---|---|---|---|---|---|
| | | Aircraft | SAT | DTD | SUN | UCF | Avg | Cars | Net | Flowers | Food | Pets | Caltech | Avg |
| **0** | CLIP 2021 | 24.8 | 47.8 | 43.8 | 62.5 | 66.7 | 47.6 | 65.5 | 66.7 | 67.4 | 85.3 | 89.1 | 92.9 | 77.7 |
| **1** | CoOp 2022b | 20.8 | 56.4 | 50.1 | 67.0 | 71.2 | 53.1 | 67.5 | 65.7 | 78.3 | 84.3 | 90.2 | 92.5 | 79.8 |
| | CoCoOp 2022a | 28.1 | 55.4 | 52.6 | 68.7 | 70.4 | 55.0 | 67.6 | 69.4 | 73.4 | 84.9 | 91.9 | 94.1 | 80.2 |
| | TIP-Adapter 2022 | 28.8 | 67.8 | 51.6 | 67.2 | 73.4 | 57.8 | 67.1 | 69.4 | 83.8 | 85.8 | 90.6 | 94.0 | 81.8 |
| | CLIP-Adapter 2024 | 25.2 | 49.3 | 44.2 | 65.4 | 66.9 | 50.2 | 65.7 | 67.9 | 71.3 | 86.1 | 89.0 | 92.0 | 78.7 |
| | PLOT++ 2022a | 28.6 | 65.4 | 54.6 | 66.8 | 74.3 | 58.0 | 68.8 | 66.5 | 80.5 | 86.2 | 91.9 | 94.3 | 81.4 |
| | KgCoOp 2023 | 26.8 | 61.9 | 52.7 | 68.4 | 72.8 | 56.5 | 66.7 | 68.9 | 74.7 | **86.4** | **92.1** | 94.2 | 80.5 |
| | ProGrad 2023 | 28.9 | 57.0 | 52.8 | 67.0 | 73.3 | 55.8 | 68.2 | 67.0 | 80.9 | 84.9 | 91.4 | 93.5 | 81.0 |
| | CLIP-LoRA 2024 | 28.0 | 71.9 | 54.1 | 70.3 | 75.4 | 59.9 | 69.4 | **70.3** | 81.4 | 85.1 | 91.9 | 93.8 | 82.0 |
| | +FMA 2025 | 28.3 | 73.0 | 55.1 | 70.6 | 75.9 | 60.6+0.7 | 69.8 | 70.2 | 84.9 | 85.2 | **92.1** | 94.5 | 82.8+0.8 |
| | **+HFM (Ours)** | **32.2** | **81.0** | **58.6** | **71.4** | **77.4** | **64.1**+4.2 | **70.3** | 70.0 | **86.2** | 85.4 | 92.0 | **95.2** | **83.2**+1.2 |
| **4** | CoOp 2022b | 30.9 | 69.7 | 59.5 | 69.7 | 77.6 | 61.5 | 74.4 | 68.8 | 92.2 | 84.5 | 92.5 | 94.5 | 84.5 |
| | CoCoOp 2022a | 30.6 | 61.7 | 55.7 | 70.4 | 75.3 | 58.7 | 69.5 | 70.6 | 81.5 | 86.3 | 92.7 | 94.8 | 82.6 |
| | TIP-Adapter 2022 | 35.7 | 76.8 | 59.8 | 70.8 | 78.1 | 64.2 | 74.1 | 70.7 | 92.1 | 86.5 | 91.9 | 94.8 | 85.0 |
| | CLIP-Adapter 2024 | 27.9 | 51.2 | 46.1 | 68.0 | 70.6 | 52.8 | 67.5 | 68.6 | 73.1 | 86.5 | 90.8 | 94.0 | 80.1 |
| | PLOT++ 2022a | 35.3 | 83.2 | 62.4 | 71.7 | 79.8 | 66.5 | 76.3 | 70.4 | 92.9 | 86.5 | 92.7 | 95.1 | 85.6 |
| | KgCoOp 2023 | 32.2 | 71.8 | 58.7 | 71.5 | 77.6 | 62.4 | 69.5 | 69.9 | 87.0 | **86.9** | 92.6 | 95.0 | 83.5 |
| | ProGrad 2024 | 34.1 | 69.6 | 59.7 | 71.7 | 77.9 | 62.6 | 75.0 | 70.2 | 91.1 | 85.4 | 92.1 | 94.4 | 84.7 |
| | CLIP-LoRA 2024 | 38.8 | 83.5 | 64.0 | 72.8 | 81.1 | 68.0 | 77.4 | 71.4 | 92.9 | 82.6 | 90.6 | 95.0 | 85.0 |
| | +FMA 2025 | 40.3 | 85.0 | 67.0 | 73.7 | 82.4 | 69.7+1.7 | 78.9 | 72.0 | 95.0 | 83.2 | 90.8 | 95.8 | 86.0+1.0 |
| | **+HFM (Ours)** | **43.5** | **90.3** | **68.6** | **75.5** | **83.6** | **72.3**+4.3 | **80.1** | **72.1** | **96.2** | **86.9** | **93.2** | **96.2** | **87.4**+2.4 |
| **16** | CoOp 2022b | 43.3 | 86.0 | 70.0 | 74.9 | 83.1 | 71.4 | 83.1 | 71.4 | 97.2 | 84.4 | 91.1 | 95.5 | 87.1 |
| | CoCoOp 2022a | 33.8 | 75.5 | 65.8 | 72.8 | 76.0 | 64.8 | 72.4 | 71.1 | 87.1 | 87.4 | 93.2 | 95.2 | 84.4 |
| | TIP-Adapter 2022 | 44.6 | 85.9 | 70.8 | 76.0 | 83.9 | 72.2 | 82.3 | 73.4 | 96.2 | 86.8 | 92.6 | 95.7 | 87.8 |
| | CLIP-Adapter 2024 | 34.2 | 71.4 | 59.4 | 74.2 | 80.2 | 63.9 | 74.0 | 69.8 | 92.9 | 87.1 | 92.3 | 94.9 | 85.2 |
| | PLOT++ 2022a | 46.7 | 92.0 | 71.4 | 76.0 | 85.3 | 74.3 | 84.6 | 72.6 | 97.6 | 87.1 | 93.6 | 96.0 | 88.6 |
| | KgCoOp 2023 | 36.5 | 76.2 | 68.7 | 73.3 | 81.7 | 67.3 | 74.8 | 70.4 | 93.4 | **87.2** | 93.2 | 95.2 | 85.7 |
| | ProGrad 2024 | 43.0 | 83.6 | 68.8 | 75.1 | 82.7 | 70.6 | 82.9 | 72.1 | 96.6 | 85.8 | 92.8 | 95.9 | 87.7 |
| | CLIP-LoRA 2024 | 54.7 | 90.7 | 73.0 | 76.0 | 86.2 | 76.1 | 86.0 | 73.4 | 97.9 | 84.2 | 91.6 | 96.1 | 88.2 |
| | +FMA 2025 | 57.8 | 91.0 | 75.4 | 77.2 | 87.1 | 77.7+1.6 | 87.7 | 73.5 | **99.1** | 85.1 | 91.6 | 96.5 | 88.9+0.7 |
| | **+HFM (ours)** | **62.1** | **94.3** | **76.0** | **77.5** | **88.9** | **79.8**+3.7 | **88.6** | **73.6** | 98.7 | **87.2** | **94.1** | **96.8** | **89.8**+1.6 |

ceeding FMA by **3.5%** and **2.1%**, respectively. This advantage is most pronounced on structurally complex datasets like EuroSAT and DTD, where HFM outperforms FMA by **8.0%** and **3.5%** under the 1-shot setting. Furthermore, compared to the strong CLIP-LoRA (Zanella & Ben Ayed, 2024) baseline, HFM yields consistent gains of **3.7%**–**4.3%** on difficult datasets, demonstrating the superiority of continuous flow adaptation over static parameter tuning.

### 4.2. Diagnostic Experiments

To verify the efficacy of HFM and each module, we performed extensive diagnostic experiments.

**Key Component Analysis.** Contributions of Centripetal Hyperbolic Alignment (CHA), flowing matching with Path-decoupled Objective (PO), and inference with Diameter-based Stopping (DS) were evaluated under both 4-shot and 16-shot settings, as summarized in Table 2. The first row in each block refers to the baseline model (*i.e.*, CLIP-LoRA (Zanella & Ben Ayed, 2024)) implemented in Euclidean space. Taking the 16-shot setting as an example, four crucial conclusions can be drawn. **First**, by simply restructuring the latent space with CHA, consistent improvements are observed, with notable gains on the challenging Aircraft benchmark (from 54.7% to **56.9%**). This demonstrates that the concentric hierarchy, *i.e.*, anchoring text as roots and images as leaves, provides a superior geometric initialization than the unstructured Euclidean space. **Second**, with the guidance of the path-decoupled objective in flow matching, the model effectively learns to decouple transport trajectories, resulting in a substantial performance leap (*e.g.*, **61.4%** on Aircraft). This indicates that the "semantic guardrail" mechanism successfully reduces inter-class interference by confining features to isolated geodesic corridors. **Third**, benefiting from the adaptive diameter-based stopping, the inference process terminates dynamically based on the intrinsic semantic scale. This yields further improvements (*e.g.*, **62.1%** on Aircraft), confirming that preventing over-transportation into the crowded origin is essential for preserving discriminability. **Finally**, this progressive im-

*Table 2.* **Ablation Study.** Analysis of component effectiveness (§4.2) under 4/16-shot settings. **CHA**: Centripetal Hyperbolic Alignment to construct the centripetal hierarchy, **PO**: flow matching with the Path-decoupled Objective, **DS**: inference with Diameter-based Stopping.

| Shots | Components | | | Difficult Datasets | | | | | | Easy Datasets | | | | | | |
|---|---|---|---|---|---|---|---|---|---|---|---|---|---|---|---|---|
| | CHA | PO | DS | FGVC | EuroSAT | DTD | SUN | UCF | Avg | Cars | Net | Flowers | Food | Pets | Caltech | Avg |
| **4** | ✗ | ✗ | ✗ | 38.8 | 83.5 | 64.0 | 72.8 | 81.1 | 68.0 | 77.4 | 71.4 | 92.9 | 82.6 | 90.6 | 95.0 | 85.0 |
| | ✓ | ✗ | ✗ | 38.9 | 88.4 | 65.9 | 74.0 | 81.8 | 69.8 | 77.6 | 71.4 | 94.4 | 86.7 | 92.6 | 95.7 | 86.4 |
| | ✓ | ✓ | ✗ | 43.5 | 88.6 | 68.0 | 75.4 | 83.6 | 71.8 | 80.5 | 71.9 | 96.4 | 86.8 | 92.7 | 95.8 | 87.3 |
| | ✓ | ✓ | ✓ | 43.5 | 90.3 | 68.6 | 75.5 | 83.6 | 72.3 | 80.1 | 72.1 | 96.2 | 86.9 | 93.2 | 96.2 | 87.4 |
| **16** | ✗ | ✗ | ✗ | 54.7 | 90.7 | 73.0 | 76.0 | 86.2 | 76.1 | 86.0 | 73.4 | 97.9 | 84.2 | 91.6 | 96.1 | 88.2 |
| | ✓ | ✗ | ✗ | 56.9 | 93.2 | 74.1 | 76.9 | 87.6 | 77.7 | 86.6 | 73.6 | 98.4 | 87.3 | 93.5 | 96.4 | 89.3 |
| | ✓ | ✓ | ✗ | 61.4 | 93.2 | 76.4 | 77.5 | 87.3 | 79.2 | 88.5 | 74.3 | 98.5 | 87.4 | 93.7 | 96.5 | 89.8 |
| | ✓ | ✓ | ✓ | 62.1 | 94.3 | 76.0 | 77.5 | 88.9 | 79.8 | 88.6 | 73.6 | 98.7 | 87.2 | 94.1 | 96.8 | 89.8 |

*Table 3.* **Ablation Study.** Analysis of model-agnostic capability (§4.2) across various PEFT approaches under the 16-shot setting.

| Method | Difficult | | | | | | Easy | | | | | | |
|---|---|---|---|---|---|---|---|---|---|---|---|---|---|
| | Aircraft | SAT | DTD | SUN | UCF | Avg | Cars | Net | Flowers | Food | Pets | Caltech | Avg |
| CoOp 2022b | 43.3 | 86.0 | 70.0 | 74.9 | 83.1 | 71.4 | 83.1 | 71.4 | 97.2 | 84.4 | 91.1 | 95.5 | 87.1 |
| +FMA 2025 | 47.6 | 88.1 | 73.1 | 75.9 | 84.4 | 73.8$_{+2.4}$ | 85.4 | 72.5 | 98.2 | 85.0 | 91.4 | 95.7 | 88.0$_{+0.9}$ |
| **+HFM (ours)** | 49.2 | 88.9 | 75.1 | 76.4 | 84.7 | 74.9$_{+3.5}$ | 84.9 | 72.2 | 98.2 | 86.6 | 93.8 | 96.3 | 88.7$_{+1.6}$ |
| CoCoOp 2022a | 33.8 | 75.5 | 65.8 | 72.8 | 76.0 | 64.8 | 72.4 | 71.1 | 87.1 | 87.4 | 93.2 | 95.2 | 84.4 |
| +FMA 2025 | 36.9 | 86.9 | 71.9 | 73.4 | 80.3 | 69.9$_{+5.1}$ | 73.5 | 71.9 | 94.5 | 87.8 | 93.4 | 95.6 | 86.1$_{+1.7}$ |
| **+HFM (ours)** | 43.7 | 89.3 | 72.6 | 75.6 | 83.9 | 73.0$_{+8.2}$ | 79.3 | 71.9 | 97.9 | 87.1 | 93.9 | 96.3 | 87.7$_{+3.3}$ |
| CLIP-Adapter 2024 | 33.8 | 70.4 | 59.3 | 74.3 | 80.1 | 63.6 | 74.2 | 71.6 | 93.6 | 87.1 | 92.4 | 94.9 | 85.6 |
| +FMA 2025 | 35.8 | 85.6 | 69.2 | 74.4 | 81.5 | 69.3$_{+5.7}$ | 74.7 | 71.3 | 95.6 | 87.2 | 92.9 | 96.0 | 86.3$_{+0.7}$ |
| **+HFM (ours)** | 43.8 | 85.7 | 71.4 | 76.1 | 84.6 | 72.3$_{+8.7}$ | 80.6 | 72.1 | 97.2 | 87.0 | 93.3 | 96.2 | 87.7$_{+2.1}$ |
| CLIP-LoRA 2024 | 54.7 | 90.7 | 73.0 | 76.0 | 86.2 | 76.1 | 86.0 | 73.4 | 97.9 | 84.2 | 91.6 | 96.1 | 88.2 |
| +FMA 2025 | 57.8 | 91.0 | 75.4 | 77.2 | 87.1 | 77.7$_{+1.6}$ | 87.7 | 73.5 | 99.1 | 85.1 | 91.6 | 96.5 | 88.9$_{+0.7}$ |
| **+HFM (ours)** | 62.1 | 94.3 | 76.0 | 77.5 | 88.9 | 79.8$_{+3.7}$ | 88.6 | 73.6 | 98.7 | 87.2 | 94.1 | 96.8 | 89.8$_{+1.6}$ |

provement is consistent across different data settings. For the difficult datasets, the average accuracy increases from 68.0% to **72.3**% in 4-shot setting, paralleling the trend in the 16-shot setting (76.1% to **79.8**%). This consistency verifies that HFM is robust to data scarcity and can effectively resolve path-entanglement regardless of the support set size.

**Model-Agnostic Generalization.** We investigated the generalizability of HFM by integrating it with various PEFT architectures, as shown in Table 3. As a plug-and-play module, HFM consistently yields superior performance when applied to CoOp (Zhou et al., 2022b), CoCoOp (Zhou et al., 2022a), CLIP-Adapter (Gao et al., 2024), and CLIP-LoRA (Zanella & Ben Ayed, 2024). For instance, when equipping CLIP-Adapter (Gao et al., 2024) with HFM, the average accuracy on difficult datasets yields a remarkable surge of **8.7**% (from 63.6% to 72.3%), significantly outperforming the **5.7**% gain obtained by the Euclidean counterpart FMA (Jiang et al., 2025). Similarly, on CoCoOp (Zhou et al., 2022a), HFM achieves an **8.2**% improvement, surpassing FMA by a clear margin of **3.1**%. Even on the stronger CoOp (Zhou et al., 2022b) baseline, HFM provides a consistent **3.5**% gain on difficult benchmarks. These results confirm that the benefit of hyperbolic geometry is orthogonal to the choice of parameter-tuning strategy, serving as a universal geometric enhancer for resolving path-entanglement.

**Scalability across Backbones.** Table 4 reports the performance using different CLIP backbones (ViT-B/32, ViT-B/16, and ViT-L/14) under both 4/16-shot settings. HFM consistently enhances the baseline CLIP-LoRA across all model sizes. Specifically, on the difficult benchmarks, HFM achieves remarkable accuracy gains. For instance, with the ViT-B/32 backbone, our method improves performance by **2.6**% (72.3% *vs.* 74.9%) in the 16-shot setting. This advantage scales effectively to the larger ViT-L/14 model, where HFM yields consistent improvements of **2.1**% (81.0% *vs.* 83.1%) on difficult datasets, while reaching a peak average accuracy of **93.1**% on the easy group. Furthermore, in the data-scarce 4-shot setup, HFM boosts the widely used ViT-B/16 backbone by a substantial margin of **4.3**% on difficult groups. This demonstrates that our method scales effectively with model capacity and feature dimensionality, maintaining its superiority even with stronger visual encoders.

### 4.3. Qualitative Comparison

To intuitively understand how HFM resolves path entanglement, we visualized the transport trajectories on five difficult benchmarks using PCA. As illustrated in Figure 3, we can observe that: 1) **Chaotic Crossovers in Euclidean Flows.** The top row reveals that Euclidean flow matching suffers from severe *path entanglement*. Due to the limited spatial

*Table 4.* **Ablation study.** Performance comparison (§4.2) between CLIP-LoRA and our HFM on different backbones (ViT-B32, ViT-B16, and ViT-L14) with 4 and 16 shots. Top-1 accuracy is reported. The highest value of each setting is bolded.

| Shots | Backbone | Method | Difficult | | | | | | Easy | | | | | | |
|---|---|---|---|---|---|---|---|---|---|---|---|---|---|---|---|
| | | | Aircraft | SAT | DTD | SUN | UCF | Avg | Cars | Net | Flowers | Food | Pets | Caltech | Avg |
| 4 | ViT-B32 | CLIP-LoRA 2024 | 27.7 | 85.6 | 60.3 | 70.3 | 76.5 | 64.1 | 68.3 | 66.5 | 90.1 | 75.6 | 86.3 | 94.3 | 80.2 |
| | | +HFM (ours) | **33.9** | **90.1** | **64.5** | **72.7** | **80.1** | **68.3**+4.2 | **71.6** | **66.7** | **93.1** | **80.7** | **90.6** | **95.4** | **83.0**+2.8 |
| | ViT-B16 | CLIP-LoRA 2024 | 38.8 | 83.5 | 64.0 | 72.8 | 81.1 | 68.0 | 77.4 | 71.4 | 92.9 | 82.6 | 90.6 | 95.0 | 85.0 |
| | | +HFM (ours) | **43.5** | **90.3** | **68.6** | **75.5** | **83.6** | **72.3**+4.3 | **80.1** | **72.1** | **96.2** | **86.9** | **93.2** | **96.2** | **87.4**+2.4 |
| | ViT-L14 | CLIP-LoRA 2024 | 48.9 | 86.4 | 70.4 | 76.7 | 86.4 | 73.8 | 85.2 | 77.9 | 97.4 | 89.6 | 93.9 | 97.2 | 90.2 |
| | | +HFM (ours) | **54.8** | **88.6** | **72.2** | **78.9** | **88.3** | **76.6**+2.8 | **87.0** | **78.9** | **98.7** | **91.8** | **95.3** | **97.7** | **91.6**+1.4 |
| 16 | ViT-B32 | CLIP-LoRA 2024 | 44.9 | 91.8 | 68.2 | 74.0 | 82.8 | 72.3 | 79.7 | 68.4 | 96.2 | 78.2 | 88.8 | 95.2 | 84.4 |
| | | +HFM (ours) | **50.4** | **93.1** | **71.0** | **75.0** | **85.2** | **74.9**+2.6 | **81.8** | **68.5** | **97.3** | **81.0** | **89.3** | **95.9** | **85.6**+1.2 |
| | ViT-B16 | CLIP-LoRA 2024 | 54.7 | 90.7 | 73.0 | 76.0 | 86.2 | 76.1 | 86.0 | 73.4 | 97.9 | 84.2 | 91.6 | 96.1 | 88.2 |
| | | +HFM (ours) | **62.1** | **94.3** | **76.0** | **77.5** | **88.9** | **79.8**+3.7 | **88.6** | **73.6** | **98.7** | **87.3** | **94.1** | **96.8** | **89.8**+1.6 |
| | ViT-L14 | CLIP-LoRA 2024 | 66.2 | 93.1 | 76.5 | 79.4 | 89.9 | 81.0 | 90.9 | 79.6 | 99.0 | 89.9 | 94.3 | 97.3 | 91.8 |
| | | +HFM (ours) | **68.7** | **93.4** | **80.1** | **81.6** | **91.5** | **83.1**+2.1 | **92.0** | **80.7** | **99.3** | **92.2** | **96.2** | **98.0** | **93.1**+1.3 |

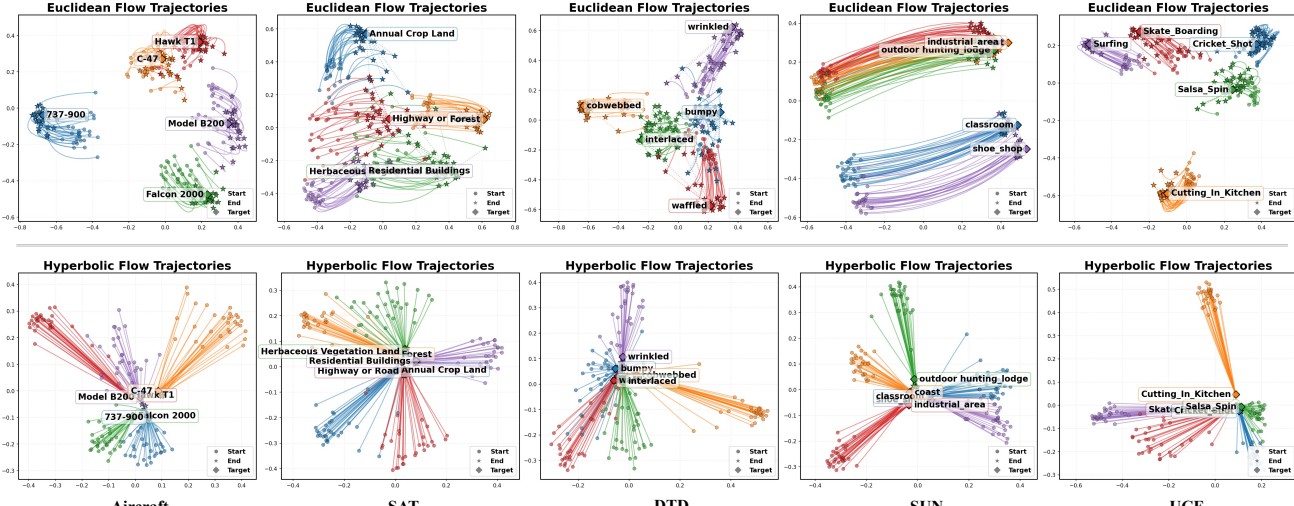

*Figure 3.* **Qualitative Results.** Visualization of transport trajectories (§4.3). **Top:** Euclidean flows suffer from severe *path entanglement*, exhibiting chaotic and intersecting paths due to spatial crowding. **Bottom:** HFM achieves *path decoupling* via a *centripetal hierarchy*. Visual features move centripetally from the boundary to central text roots along isolated geodesic corridors.

capacity of flat geometry, trajectories from different classes are densely packed and frequently intersect. For instance, in the Aircraft and DTD datasets, the flow paths of neighboring categories (*e.g.*, different textures or aircraft models) exhibit chaotic crossovers, leading to ambiguous decision boundaries and potential misclassification. 2) **Ordered Decoupling in Hyperbolic Flows.** In contrast, the bottom row demonstrates that HFM generates highly organized, radial trajectories. Driven by our *centripetal hyperbolic alignment*, the visual features are transported from the boundary inward to the central text roots along isolated geodesic corridors. This structure effectively exploits the exponential volume expansion of the Lorentz manifold to separate diverse semantic clusters. Consequently, inter-class collisions are virtually eliminated, confirming the efficacy of our path-decoupled objective in preserving semantic discriminability during the adaptation process.

## 5. Conclusion

In this paper, we identified that the polynomial volume growth of Euclidean space limits existing flow matching methods, leading to severe path entanglement in few-shot adaptation. To this end, we proposed path-decoupled Hyperbolic Flow Matching (HFM), which exploits the exponential expansion of the Lorentz manifold to decouple transport trajectories. Through centripetal hyperbolic alignment, a path-decoupled objective, and adaptive diameter-based stopping, HFM constructs isolated geodesic corridors and ensures precise inference termination. Empirical results on 11 benchmarks confirm that HFM not only establishes a new state-of-the-art but also demonstrates remarkable data efficiency in complex visual scenarios. We hope this work encourages further exploration into non-Euclidean generative dynamics for robust cross-modal understanding.

## Impact Statement

This paper presents work whose goal is to advance the field of machine learning. There are many potential societal consequences of our work, none of which we feel must be specifically highlighted here.

## Acknowledgements

This work was supported by the Fundamental and Interdisciplinary Disciplines Breakthrough Plan of the Ministry of Education of China (JYB2025XDXM103), National Natural Science Foundation of China (NSFC) Young Scientists Fund Category B (62522216), National Natural Science Foundation of China (NSFC) Young Scientists Fund Category C (62402408), Hong Kong SAR Research Grants Council (RGC) Early Career Scheme (26208924), Hong Kong SAR Research Grants Council (RGC) General Research Fund (16219025) and the Hong Kong SAR RGC General Research Fund under Grant (16208823). This research was partially conducted by ACCESS – AI Chip Center for Emerging Smart Systems, supported by the InnoHK initiative of the Innovation and Technology Commission of the Hong Kong Special Administrative Region Government. This work was also supported by the Key R&D Program of Zhejiang (2025C01128), the National Natural Science Foundation of China (62441617), Zhejiang Provincial Natural Science Foundation of China (No. LD25F020001) and Fundamental Research Funds for the Central Universities (226-2025-00057).

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

## Appendix Overview

This appendix is organized as follows:

- §A.1 provides a quantitative evaluation of path entanglement.

- §A.2 presents fine-grained ablation studies.

- §A.3 analyzes the necessity of hyperbolic geometry.

- §A.4 studies the necessity of step-wise consistency supervision.

- §A.5 reports the time and memory overhead of HFM.

- §B discusses the limitations of HFM.

## A. Additional Experimental Analyses

### A.1. Quantitative Evaluation of Path Entanglement

Although path entanglement is inherently difficult to quantify with a single scalar metric, we introduce a trajectory-level empirical proxy, termed *Trajectory Entanglement Rate* (TER), to assess whether a transported feature becomes closer to an incorrect prototype along its trajectory. For a trajectory at step $t$, let $d_t^+$ denote the geodesic distance to the ground-truth prototype and $d_t^-$ denote the distance to the nearest incorrect prototype. We define the normalized margin as

$$m_t = \frac{d_t^- - d_t^+}{d_t^- + d_t^+ + \epsilon}. \tag{11}$$

An entanglement event is identified when $m_t < 0$, indicating that the transported feature is closer to an incorrect prototype than to its ground-truth prototype. TER is then computed as the fraction of samples whose trajectories contain at least one such event. A lower TER suggests that the learned transport paths are less likely to drift into competing semantic regions.

Table 5 reports the TER results on five difficult benchmarks. Compared with FMA, HFM consistently reduces TER across all datasets, decreasing the average TER from 30.2% to 27.8%. This result provides quantitative support for our claim that hyperbolic transport helps alleviate trajectory entanglement, complementing the qualitative trajectory visualizations in the main paper.

### A.2. Fine-grained Ablations

We provide fine-grained ablations for the two main training components of HFM: Centripetal Hyperbolic Alignment (CHA) and the Path-decoupled Objective (PO). All experiments are conducted under the 16-shot setting on the five difficult benchmarks.

*Table 5.* **Trajectory entanglement rate comparison.** Lower TER indicates less entangled transport trajectories.

| Method | Aircraft | SAT | DTD | SUN | UCF | Avg |
|--------|----------|------|------|------|------|------|
| FMA | 55.8 | 10.6 | 32.4 | 33.9 | 18.1 | 30.2 |
| HFM | 52.5 | 9.1 | 30.7 | 30.4 | 16.4 | 27.8 |

*Table 6.* **Ablation study on CHA.** Fine-grained analysis of the entailment and contrastive losses under the 16-shot setting.

| $\mathcal{L}_{entail}$ | $\mathcal{L}_{con}$ | Aircraft | SAT | DTD | SUN | UCF | Avg |
|--------|--------|----------|------|------|------|------|------|
| ✓ | ✗ | 61.5 | 92.8 | 75.8 | 76.9 | 88.1 | 79.0 |
| ✗ | ✓ | 44.6 | 89.5 | 71.9 | 70.8 | 83.2 | 72.0 |
| ✓ | ✓ | 62.1 | 94.3 | 76.0 | 77.5 | 88.9 | 79.8 |

*Table 7.* **Ablation study on PO.** Fine-grained analysis of the step-wise consistency and inter-class decoupling losses under the 16-shot setting.

| $\mathcal{L}_{step}$ | $\mathcal{L}_{icd}$ | Aircraft | SAT | DTD | SUN | UCF | Avg |
|--------|--------|----------|------|------|------|------|------|
| ✓ | ✗ | 61.4 | 93.8 | 75.6 | 77.7 | 88.7 | 79.4 |
| ✗ | ✓ | 60.8 | 92.9 | 75.6 | 77.2 | 87.3 | 78.8 |
| ✓ | ✓ | 62.1 | 94.3 | 76.0 | 77.5 | 88.9 | 79.8 |

**Ablation on CHA.** Table 6 analyzes the two losses used in CHA. Using only the entailment loss $\mathcal{L}_{entail}$ already yields a strong average accuracy of 79.0%, suggesting that enforcing the text-to-image partial order is important for establishing the centripetal hierarchy. In contrast, using only the contrastive loss $\mathcal{L}_{con}$ leads to a much lower average accuracy of 72.0%, indicating that class discrimination alone is insufficient to construct the desired hyperbolic transport geometry. Combining both losses achieves the best average accuracy of 79.8%, showing that semantic discrimination and hierarchical ordering are complementary.

**Ablation on PO.** Table 7 studies the two terms in the Path-decoupled Objective. When $\mathcal{L}_{step}$ is disabled, we replace it with the standard velocity supervision, while keeping $\mathcal{L}_{icd}$ unchanged. The step-wise consistency loss $\mathcal{L}_{step}$ provides strong supervision by directly constraining the predicted next state on the manifold, achieving 79.4% average accuracy. The inter-class decoupling loss $\mathcal{L}_{icd}$ also performs competitively, but is slightly weaker when used alone. Combining the two terms gives the best result of 79.8%, suggesting that $\mathcal{L}_{step}$ stabilizes the geometric trajectory while $\mathcal{L}_{icd}$ further discourages drift toward incorrect semantic regions.

### A.3. Necessity of Hyperbolic Geometry

To disentangle the effect of objective design from the effect of geometry, we conduct a Euclidean-space control experiment, denoted as FMA*. FMA* applies the same contrastive and step-wise consistency objectives used in HFM, but keeps the transport space Euclidean. As shown in Table 9, FMA* only slightly improves over FMA, in-

*Table 8.* **Computational overhead comparison.** We compare HFM with the Euclidean FMA baseline under a capacity-matched setting.

| Method | Params (M) | Param Mem (MB) | Train / step (ms) | Peak Train Mem (MB) | Infer (ms) | Peak Infer Mem (MB) |
|---|---|---|---|---|---|---|
| FMA | 17.22 | 65.68 | 9.288 | 392.70 | 0.622 | 197.30 |
| HFM | 17.28 | 65.93 | 11.774 | 409.87 | 0.739 | 208.92 |

*Table 9.* **Euclidean-space control experiment.** FMA* applies the same objectives as HFM in Euclidean space.

| Method | Aircraft | SAT | DTD | SUN | UCF | Avg |
|---|---|---|---|---|---|---|
| FMA | 57.8 | 91.0 | 75.4 | 77.2 | 87.1 | 77.7 |
| FMA* | 57.8 | 90.9 | 76.3 | 77.4 | 86.9 | 77.9 |
| HFM | 62.1 | 94.3 | 76.0 | 77.5 | 88.9 | 79.8 |

*Table 10.* **Velocity supervision vs. step-wise consistency.** We compare conventional velocity supervision with the proposed step-wise consistency supervision.

| Model | Supervision | Aircraft | SAT | DTD | SUN | UCF | Avg |
|---|---|---|---|---|---|---|---|
| FMA | Velocity | 57.8 | 91.0 | 75.4 | 77.2 | 87.1 | 77.7 |
| FMA | Step | 57.9 | 90.6 | 76.2 | 77.3 | 87.2 | 77.8 |
| HFM | Velocity | 60.8 | 92.9 | 75.6 | 77.2 | 87.3 | 78.8 |
| HFM | Step | 62.1 | 94.3 | 76.0 | 77.5 | 88.9 | 79.8 |

creasing the average accuracy from 77.7% to 77.9%. In contrast, HFM achieves 79.8%, outperforming FMA* by 1.9%. These results suggest that the performance gain of HFM cannot be attributed to the objectives alone; the hyperbolic geometry itself plays a central role in improving trajectory decoupling.

### A.4. Necessity of Step-wise Consistency

We further compare conventional velocity supervision with our step-wise consistency supervision. As shown in Table 10, replacing velocity supervision with step-wise consistency brings only a marginal gain for Euclidean FMA (Jiang et al., 2025), improving the average accuracy from 77.7% to 77.8%. This is expected because Euclidean flow paths are straight and direction-based velocity supervision already provides a direct learning signal. In contrast, step-wise consistency is more beneficial in HFM, improving the average accuracy from 78.8% to 79.8%. This indicates that directly supervising the predicted next state on the hyperbolic manifold better regularizes the multi-step transport process, reducing accumulated trajectory drift and helping maintain class-specific geodesic corridors.

### A.5. Computational Overhead

We further analyze the computational overhead of HFM compared with the Euclidean FMA (Jiang et al., 2025) baseline under a capacity-matched setting. Specifically, both methods use the same flow network, 512-dimensional features, a batch size of 32, 10 inference steps, and the same

hardware environment. As shown in Table 8, HFM introduces only marginal parameter overhead, increasing the number of parameters from 17.22M to 17.28M. The memory overhead is also modest, with peak training and inference memory increasing by approximately 4.4% and 5.9%, respectively. The main additional cost comes from hyperbolic geometric operations, including logarithmic/exponential maps, tangent-space projection, and geodesic distance computation. As a result, HFM incurs a moderate runtime increase during both training and inference. Nevertheless, the overhead remains controlled relative to the performance gains reported in the main paper.

## B. Limitations

Despite its effectiveness, HFM still has several limitations. First, as a multi-step flow-based adaptation method, HFM incurs higher inference latency than one-step feature adapters. Although the proposed diameter-based stopping strategy helps reduce unnecessary transport, the overall inference cost remains larger than that of single-pass adaptation methods. Second, our experiments mainly focus on few-shot image classification with vision-language models. Extending hyperbolic flow matching to broader multimodal perception tasks, such as dense prediction, retrieval, or visual reasoning, may require task-specific trajectory design and supervision objectives, which we leave for future work.

