# OpenReview forum: "Path-Decoupled Hyperbolic Flow Matching for Few-Shot Adaptation"
_ICML.cc/2026/Conference — ICML 2026 regular_

### Official Review · Reviewer_bTtr · 2026-02-24

**Soundness:** 3
**Presentation:** 4
**Significance:** 3
**Originality:** 3
**Overall Recommendation:** 5
**Confidence:** 4

**Summary:**

This paper tackles few-shot adaptation for VLM-based image classification by transporting image features toward class text prototypes via a continuous-time flow. The authors argue that Euclidean Flow Matching (EFM) can suffer from *path entanglement*, where class-wise transport trajectories interfere and cause misclassification, and propose Hyperbolic Flow Matching (HFM) on the Lorentz manifold to better separate trajectories. The method consists of (i) Centripetal Hyperbolic Alignment (CHA) to place text prototypes near the center and image features near the boundary, (ii) a Path-decoupled Objective (PO) combining a step-wise consistency loss and an inter-class decoupling loss, and (iii) Diameter-based Stopping (DS) to avoid overly entering crowded central regions during inference. Experiments on multiple few-shot benchmarks demonstrate consistent improvements over several EFM-based and PEFT baselines.

**Compliance With Llm Reviewing Policy:**

Affirmed.

**Final Justification:**

I update my recommendation from **4 (Weak Accept)** to **5 (Accept)**.

My main concerns in the original review were all directly tied to the paper’s central claim: whether the improvement should really be attributed to hyperbolic geometry and path decoupling, rather than to auxiliary objectives or loosely motivated design choices. In particular, I asked for (1) fine-grained ablations within CHA and PO, (2) a Euclidean control with matched objectives, (3) a direct comparison between velocity supervision and the proposed step-wise consistency loss, and (4) a quantitative evaluation of path entanglement.

The rebuttal addressed all of these concerns with targeted additional experiments. The fine-grained ablations show that removing each component hurts performance. The Euclidean matched-objective control improves only marginally over FMA, while HFM remains clearly stronger, which substantially strengthens the claim that hyperbolic geometry itself is important rather than merely the added objectives. The velocity-vs-step-consistency comparison also clarifies that the proposed step-wise consistency loss is meaningfully beneficial in the hyperbolic setting. Finally, the proposed TER metric provides useful quantitative support for the path-entanglement hypothesis that motivates the paper.

After the rebuttal, I find the paper’s main empirical and conceptual claims to be sufficiently supported. The paper is well presented, addresses an important problem in few-shot VLM adaptation, and the additional evidence resolves my main soundness concerns. For these reasons, I now view the paper more positively and raise my recommendation by one level.

**Key Questions For Authors:**

1. Euclidean controls with matched objectives:

- Can you run Euclidean-space controls that mirror the proposed objectives as closely as possible, and report how much gap remains to HFM?

    - If Euclidean controls yield clearly smaller gains and HFM remains strongly superior, it would substantially strengthen the central claim and would increase the overall score.

    - If Euclidean controls achieve similar gains, my evaluation would shift toward interpreting the contribution primarily as the objective/regularization design rather than hyperbolic geometry; this would weaken the “hyperbolic necessity” claim and could reduce the overall score.

2. Fine-grained necessity ablations for CHA and PO

- Can you provide fine-grained ablations establishing the necessity of each element in CHA (Geometric Stratification, $\mathcal{L}\_{\text{entail}}$, $\mathcal{L}\_{\text{con}}$) and PO ($\mathcal{L}\_{\text{step}}$, $\mathcal{L}\_{\text{icd}}$)?

    - If removing any element consistently hurts performance, it would validate the proposed design as necessary and strengthen my evaluation.

    - If some elements can be removed without loss, it would suggest the method can be simplified and that parts of the current narrative/justification should be revised; this would weaken my evaluation.

3. Velocity supervision vs. $\mathcal{L}\_{\text{step}}$

- Can you directly compare classical velocity supervision versus the step-wise consistency loss $\mathcal{L}\_{\text{step}}$?

    - If the step-wise consistency loss $\mathcal{L}\_{\text{step}}$ provides clear and consistent improvements, it would strengthen the necessity claim for the step-wise consistency loss $\mathcal{L}\_{\text{step}}$ and increase my evaluation.

    - If the difference is small or inconsistent, it would weaken the necessity claim for the step-wise consistency loss $\mathcal{L}\_{\text{step}}$ and reduce my evaluation.

4. Quantitative evaluation of “path entanglement”

- Can you provide quantitative evidence supporting the “path entanglement” hypothesis (subset evaluations and/or explicit entanglement metrics)?

    - If quantitative metrics align with the hypothesis and correlate with accuracy improvements, the central story becomes much stronger and my evaluation would improve.

    - If such evidence is weak or inconclusive, the core motivation would appear less supported and my evaluation would decrease.

**Limitations:**

No. The paper includes an impact statement regarding societal considerations, but technical limitations are not clearly articulated. A brief discussion of method-specific limitations would improve transparency and reproducibility.

**Strengths And Weaknesses:**

# Strengths

## Soundness

- The paper clearly frames path entanglement as a central failure mode and motivates hyperbolic geometry as a means to separate transport corridors; the overall story is coherent.

- The method is organized into three major components (CHA/PO/DS), and the paper includes ablations at this coarse granularity demonstrating additive gains.

- The method is shown to be plug-and-play across multiple PEFT approaches, supporting broad applicability rather than a single tightly-coupled pipeline.

## Presentation

- The paper is well-structured from the Lorentz model preliminaries to the formalization of each component, and the figures are clear and helpful for building intuition.

## Significance

- Few-shot adaptation for VLMs is an important practical problem, and the paper demonstrates consistent improvements across many benchmarks.

##  Originality

- While hyperbolic representations are not entirely new, the paper’s novelty lies in positioning trajectory interference as the key issue in few-shot adaptation and presenting an integrated framework for few-shot VLM adaptation that combines CHA, PO, and DS to address it.

# Weaknesses

## Soundness

- **Lack of fine-grained ablations**: The current ablations are mostly at the block level (CHA/PO/DS). The paper does not provide fine-grained ablations for Geometric Stratification, $\mathcal{L}\_{\text{entail}}$, $\mathcal{L}\_{\text{con}}$, $\mathcal{L}\_{\text{step}}$, and $\mathcal{L}\_{\text{icd}}$. As a result, it is difficult to assess the necessity of each component.

- **Necessity of hyperbolic geometry**: It is not cleanly separated whether the gains come from hyperbolic geometry itself or from the additional losses/objectives. Several ingredients appear, at least conceptually, transplantable to Euclidean space. A Euclidean control with matched objectives is needed to claim that hyperbolic geometry is essential.

- **Necessity of the step-wise consistency loss $\mathcal{L}_{\text{step}}$**: Since Flow Matching is commonly understood as learning a vector field via velocity supervision, a direct comparison between classical velocity supervision and the step-wise consistency loss $\mathcal{L}_{\text{step}}$ would clarify what is fundamentally different and why the step-wise consistency loss is needed.

- **Lack of quantitative evaluation for “path entanglement”**: The paper provides qualitative evidence for “path entanglement”, but the core hypothesis would be more convincing with quantitative evaluation, e.g., performance on entanglement-prone subsets and/or metrics that quantify trajectory interference.

---

> ### Author Rebuttal · Authors · 2026-03-30
>
> ### **Q1. Fine-grained ablations.**
> We appreciate the reviewer's suggestion to conduct a more granular analysis of our modules. Per your request, we provide fine-grained ablations for each component within the **Centripetal Hyperbolic Alignment (CHA)** and the **Path-decoupled Objective (PO)** under the 16-shot setting. The results in Table R1 and Table R2 demonstrate that removing any single component hurts performance, validating our design as necessary and well-integrated. Specifically, $\mathcal{L} _ {con}$  ensures class discrimination and $\mathcal{L} _ {entail}$  further aligns the centripetal structure, while $\mathcal{L} _ {step}$ and $\mathcal{L} _ {icd}$ act synergistically to eliminate path entanglement.
>
> **Table R1. Ablation study on CHA.**
> |$\mathcal{L}_{entail}$|$\mathcal{L}_{con}$|Aircraft|SAT|DTD|SUN|UCF|Avg|
> |---|---|---|---|---|---|---|---|
> |✓||61.5|92.8|75.8|76.9|88.1|79.0|
> ||✓|44.6|89.5|71.9|70.8|83.2|72.0|
> |✓|✓|62.1|94.3|76.0|77.5|88.9|79.8|
>
> **Table R2. Ablation study on PO.**
> |$\mathcal{L}_{step}$|$\mathcal{L}_{icd}$|Aircraft|SAT|DTD|SUN|UCF|Avg|
> |---|---|---|---|---|---|---|---|
> |✓||61.4|93.8|75.6|77.7|88.7|79.4|
> ||✓|60.8|92.9|75.6|77.2|87.3|78.8|
> |✓|✓|62.1|94.3|76.0|77.5|88.9|79.8|
>
> ### **Q2. Necessity of hyperbolic geometry.**
> Thanks for the helpful comment. We conduct the Euclidean-space control experiment (FMA*) by applying the same objectives (Contrastive loss and Step-wise consistency) used in HFM. The results on 5 difficult benchmarks are as shown in Table R3. The performance gain of FMA* over the FMA is marginal (+0.2%), while HFM remains strongly superior with a significant gap (+2.1%).  It provides empirical proof that the performance leap stems from the **intrinsic properties of hyperbolic geometry** rather than objective design alone, thereby strengthening our central claim of hyperbolic necessity.
>
> **Table R3. Euclidean-space controls experiment.**
> |Method|Aircraft|SAT|DTD|SUN|UCF|Avg|
> |---|---|---|---|---|---|---|
> |FMA|57.8|91.0|75.4|77.2|87.1|77.7|
> |FMA*|57.8|90.9|76.3|77.4|86.9|77.9|
> |HFM|62.1|94.3|76.0|77.5|88.9|79.8|
>
> ### **Q3. Necessity of the step-wise consistency loss.**
> Thanks for the insightful comment. We conduct a comparison between **velocity** and our **step-wise consistency** supervision across 5 difficult benchmarks in Table R4.  $\mathcal{L}_{step}$ yields a consistent **1.0%** gain in HFM, while its Euclidean impact is negligible (+0.1%). In flat Euclidean space, **straight-line flows** make direction-based velocity supervision sufficient. However, hyperbolic geodesics are hypersensitive to **negative curvature**; *velocity errors in the tangent space amplify exponentially*, causing trajectory drift. By enforcing positional consistency on the manifold, $\mathcal{L} _ {step}$ provides the necessary rectification to confine features within class-specific corridors, resolving the path entanglement that standard velocity supervision cannot.
>
> **Table R4. Velocity supervision vs step-wise consistency.**
> |Model|Supervision|Aircraft|SAT|DTD|SUN|UCF|Avg|
> |---|---|---|---|---|---|---|---|
> |FMA|Velocity|57.8|91.0|75.4|77.2|87.1|77.7|
> |FMA|Step Consistency|57.9|90.6|76.2|77.3|87.2|77.8|
> |HFM|Velocity|60.8|92.9|75.6|77.2|87.3|78.8|
> |HFM|Step Consistency|62.1|94.3|76.0|77.5|88.9|79.8|
>
> ### **Q4. Quantitative evaluation for “path entanglement”.**
> We thank the reviewer for highlighting the necessity of a quantitative evaluation for path entanglement. While quantifying this entanglement is inherently difficult and remains an *open research question*, we try to devise the **Trajectory Entanglement Rate (TER)** metric to provide an empirical assessment. For a trajectory at step $t$, let $d^+_t$ be the distance to the ground-truth prototype and $d^-_t$ the distance to the nearest incorrect prototype. We define the normalized margin $m_t = \frac{d^-_t - d^+_t}{d^-_t + d^+_t + \epsilon}$. An entanglement event is identified if $m_t < 0$, signifying that the feature has drifted closer to an incorrect category. TER represents the fraction of samples whose trajectories experience at least one such event. We compared HFM with the Euclidean FMA baseline across 5 difficult benchmarks. A lower TER indicates a more decoupled transport path. Table R5 shows HFM consistently outperforms FMA (-2.4% TER), validating the Lorentz manifold’s exponential volume for superior trajectory decoupling.
>
> **Table R5. Trajectory Entanglement Rate (↓) Comparison.**
> |Method|Aircraft|SAT|DTD|SUN|UCF|Avg|
> |---|---|---|---|---|---|---|
> |FMA|55.8|10.6|32.4|33.9|18.1|30.2|
> |HFM|52.5|9.1|30.7|30.4|16.4|27.8|
>
> ### **Q5. Limitations.**
> We will add the *Limitations* section to the final manuscript: 1) **Inference Latency**: As a multi-step flow-based method, HFM is inherently slower than one-step feature adapters. 2) **Scope**: While our HFM works well on few-shot adaptation, extending flow matching to other multi-modality perception tasks is non-trivial and remains for future exploration.

---

> > ### Author Rebuttal · Reviewer_bTtr · 2026-04-02
> >
> > Thank you for the additional experiments and clarifications. My concerns have been fully addressed, and I will raise my score by one level.

---

> > > ### Author Response · Authors · 2026-04-02
> > >
> > > We sincerely thank the reviewer for the positive feedback and for increasing the score. We are encouraged that our responses fully addressed your concerns. We will incorporate your suggestions into the final manuscript to further improve the paper. We remain open to any further questions or suggestions you may have. Thanks again for your time and effort in reviewing our work.

---

### Official Review · Reviewer_EaGV · 2026-03-12

**Soundness:** 3
**Presentation:** 3
**Significance:** 4
**Originality:** 4
**Overall Recommendation:** 4
**Confidence:** 4

**Summary:**

This paper proposes Path-Decoupled Hyperbolic Flow Matching (HFM) for cross-modal few-shot adaptation in Vision-Language Models. It identifies the “path entanglement” issue in Euclidean Flow Matching and solves it by leveraging the exponential expansion of the Lorentz manifold. Experiments across 11 datasets and various PEFT architectures demonstrate consistent improvements over Euclidean baselines.

**Compliance With Llm Reviewing Policy:**

Affirmed.

**Final Justification:**

The author addressed my main problems and concerns, so I maintain my original score.

**Key Questions For Authors:**

Could the authors provide a quantitative comparison of training time, memory footprint, and inference latency between HFM and the Euclidean FMA baseline?
How sensitive is the model's performance to the hyperbolic curvature parameter 𝑐? Does it require dataset-specific tuning?
Does the “adaptive diameter-based stopping” mechanism require manual threshold tuning, or is it fully dynamic during inference?

**Limitations:**

Yes

**Strengths And Weaknesses:**

Strengths:
This manuscript proposes to shift from Euclidean to Hyperbolic Flow Matching to resolve trajectory entanglement, which is a highly novel and well-justified application of non-Euclidean geometry.
The evaluation spans 11 datasets, various shots (1/4/16), and multiple PEFTs/backbones. Ablations cleanly validate the proposed modules.
Motivation is exceptional. Trajectory visualizations (Figures 1 and 3) effectively demonstrate the core problem and the proposed centripetal decoupling.
This work yields meaningful performance gains in few-shot VLM adaptation (e.g., +4.2% over CLIP-LoRA on difficult datasets).
Weaknesses:
The manuscript lacks quantitative analysis of the time and memory overhead introduced by Lorentz manifold operations compared to the Euclidean baseline.
The sensitivity to the hyperbolic curvature parameter and stopping thresholds is unclear.
There seems to be lack of a quantitative metric​ to measure the degree of “path entanglement”.
Some professional terms (e.g., “entailment cone”) could be explained more plainly in the context of this work.

---

> ### Author Rebuttal · Authors · 2026-03-30
>
> ### **Q1. Time and memory overhead.**
>
> Thanks for the insightful comment. We add a quantitative efficiency comparison between HFM and the Euclidean FMA baseline under a capacity-matched setting (same flow network, 512-dim features, batch size 32, and 10 inference steps on the same hardware). Overall, results in Table R1 show that HFM introduces only marginal parameter and memory overhead, while incurring a moderate runtime increase due to the additional hyperbolic geometric operations (e.g., log/exp maps and geodesic distance computation). We will add the computational overhead evaluation in the revised manuscript.
>
>  **Table R1. Computational overhead comparison.**
>
> | Method | Params (M) | Param Mem (MB) | Train / step (ms) | Peak Train Mem (MB) | Infer (ms) | Peak Infer Mem (MB) |
> | --- | --- | --- | --- | --- | --- | --- |
> | FMA | 17.22 | 65.68 | 9.288 | 392.70 | 0.622 | 197.30 |
> | HFM | 17.28 | 65.93 | 11.774 | 409.87 | 0.739 | 208.92 |
>
> ###  **Q2. Hyperbolic curvature parameter.**
>
> We appreciate the reviewer's feedback regarding parameter sensitivity. The manifold curvature $\kappa$is not a fixed hyperparameter but is **learnable** during training (cf. Line 152). It is initialized at 1.0 and automatically optimizes to the geometry of each dataset during training. HFM’s state-of-the-art performance across 11 diverse benchmarks (achieved *without manual tuning*) directly confirms its **empirical robustness** and insensitivity to hyperparameter adjustments across varying data scales.
>
> ###  **Q3. Quantitative evaluation for “path entanglement”.**
>
> We thank the reviewer for highlighting the necessity of a quantitative evaluation for path entanglement. While quantifying this entanglement is inherently difficult and remains an *open research question*, we try to devise the **Trajectory Entanglement Rate (TER)** metric to provide an empirical assessment. For a trajectory at step $t$, let $d^+_t$ be the distance to the ground-truth prototype and $d^-_t$ the distance to the nearest incorrect prototype. We define the normalized margin $m_t = \frac{d^-_t - d^+_t}{d^-_t + d^+_t + \epsilon}$. An entanglement event is identified if $m_t < 0$, signifying that the feature has drifted closer to an incorrect category. TER represents the fraction of samples whose trajectories experience at least one such event. We compared HFM with the Euclidean FMA baseline across 5 difficult benchmarks. A lower TER indicates a more decoupled transport path. Table R2 shows HFM consistently outperforms FMA (-2.4% TER), validating the Lorentz manifold’s exponential volume for superior trajectory decoupling.
>
> **Table R2. Trajectory entanglement rate (↓) Comparison.**
>
> | Method | Aircraft | SAT | DTD | SUN | UCF | Avg |
> | --- | --- | --- | --- | --- | --- | --- |
> | FMA | 55.8 | 10.6 | 32.4 | 33.9 | 18.1 | 30.2 |
> | HFM | 52.5 | 9.1 | 30.7 | 30.4 | 16.4 | 27.8 |
>
> ### **Q4. Professional terms explanation.**
>
> We appreciate the suggestion to clarify our terminology. In the revised version, we will update **Section 3.2** to provide a more intuitive: **Entailment Cone** *denotes the region of semantic containment*. Geometrically, we treat the text prototype as the “parent” and the image feature as the “child”. By constraining the image to lie within this cone, we enforce a hierarchical relationship where the general text category logically “entails” the specific visual instance.
>
> ###  **Q5. Adaptive diameter-based stopping mechanism.**
>
> The diameter-based stopping mechanism is **data-dependent** rather than manual. The threshold is automatically scaled by the **semantic diameter** $d_{txt}$ and **class cardinality** $N$ (cf. Line 263-265).
>
> - **Data-Driven Termination**: The criterion is automatically computed from the **intrinsic semantic scale**  $d_{txt}$ of the class prototypes in each specific task.
> - **Inherent Scalability**: It utilizes a fixed, density-aware function to compensate for manifold crowding as class cardinality $N$ grows, ensuring adaptive termination without heuristic adjustments.
> - **Benchmark Consistency**: This universal formulation was applied across **all 11 benchmarks** without modification, proving its robustness across diverse few-shot scenarios.

---

> > ### Author Rebuttal · Reviewer_EaGV · 2026-04-01
> >
> > My main problems and concerns have been resolved.

---

> > > ### Author Response · Authors · 2026-04-01
> > >
> > > Thanks for your positive feedback and for the time you dedicated to reviewing our work. We are glad that our responses have resolved your main concerns. If you have any other concerns or suggestions, please do not hesitate to let us know, and we are more than happy to provide additional clarification. We sincerely appreciate the suggestions you provided, which are very helpful for improving our paper. We will incorporate your suggestions and our refinements into the final manuscript.

---

### Official Review · Reviewer_mvKY · 2026-03-13

**Soundness:** 3
**Presentation:** 3
**Significance:** 2
**Originality:** 1
**Overall Recommendation:** 4
**Confidence:** 2

**Summary:**

This paper proposes Path-Decoupled Hyperbolic Flow Matching (HFM) for few-shot adaptation of vision-language models. The authors observe that existing CLIP adaptation approaches typically rely on one-step feature transformations, which may be insufficient for aligning complex image and text representations.
To address this limitation, the paper formulates feature alignment as a continuous transport process using Flow Matching, where image features are gradually transported toward text prototypes along learned trajectories. The authors argue that performing such transport in Euclidean space may lead to path entanglement, where trajectories of different classes intersect and interfere with each other. To alleviate this issue, the proposed method performs flow-based transport in a hyperbolic feature space, whose exponentially expanding volume can help separate trajectories. The method models feature transport using Riemannian flow dynamics in hyperbolic space, implemented with exponential-map-based updates to ensure that trajectories remain on the manifold.
The proposed method is evaluated on 11 standard few-shot classification benchmarks. Experimental results show that HFM consistently improves performance when combined with existing CLIP adaptation methods, particularly on more challenging datasets.

**Compliance With Llm Reviewing Policy:**

Affirmed.

**Key Questions For Authors:**

1. Computational overhead of hyperbolic operations
Since the proposed method introduces Riemannian operations in hyperbolic space, it would be helpful to clarify the computational overhead compared to Euclidean flow matching.

**Limitations:**

1. Focus limited to few-shot classification
The experiments are restricted to few-shot classification benchmarks. It remains unclear whether the proposed approach generalizes to other multimodal tasks such as image-text retrieval or dense prediction tasks.

**Strengths And Weaknesses:**

Strengths
1. Clear motivation for continuous feature alignment
The paper provides a clear motivation that feature alignment between image and text embeddings may require more expressive transformations than one-step mappings. Modeling alignment as a continuous transport process is an intuitive and principled perspective.
2. Plug-and-play compatibility with existing methods
The proposed method can be easily integrated with several existing CLIP adaptation techniques, including CoOp, CoCoOp, CLIP-Adapter and CLIP-LoRA. This improves the practicality and flexibility of the approach.

Weaknesses
1. Limited conceptual novelty
The core idea of combining flow-based alignment with hyperbolic geometry is intuitive given prior work demonstrating the effectiveness of both techniques. As a result, the conceptual novelty of the method may be somewhat limited. Nevertheless, the paper provides a reasonable technical formulation and empirical validation of this combination.
2. Lack of comparison with existing hyperbolic VLM models
The paper does not compare against prior work that also employs hyperbolic geometry in vision–language models, such as

MERU - Hyperbolic Image-Text Representations (Desai et al., 2023)

HYPE - Hyperbolic Entailment Filtering for Underspecified Images and Texts (Kim et al., 2024)

Although these models focus on representation learning rather than adaptation, discussing or experimentally comparing them would help better position the proposed approach.

---

> ### Author Rebuttal · Authors · 2026-03-30
>
> ### **Q1. Conceptual novelty.**
> Thanks for your feedback. We would like to clarify that our work is driven by a critical, previously unaddressed discovery: **Path Entanglement** in Euclidean flows.
> - **New Problem Insight**: We demonstrate that Euclidean geometry’s polynomial growth causes transport trajectories to collide and overlap, fundamentally limiting few-shot adaptation. Identifying this “entanglement” is a novel contribution to the VLM community.
> - **Specialized Technical Design**: HFM is not a vanilla combination. We designed ***Centripetal Hyperbolic Alignment*** to structure the latent space into a root-leaf hierarchy and the ***Path-decoupled Objective*** to enforce isolated geodesic corridors. These are non-trivial adaptations specifically tailored to the Lorentz manifold's properties.
> - **Empirical Necessity**: The significant performance gap (e.g., +2.1~3.5% over Euclidean counterparts on difficult datasets) strongly proves that our method is a *necessary geometric advancement* rather than an incremental technical swap.
>
> Given the discovery of path entanglement and its targeted solution via specialized hyperbolic flow matching, our work deserves to be shared with the community. We will further clarify the relationship between our contributions and related work in revision.
>
> ### **Q2. Comparison with existing hyperbolic VLM models.**
> We appreciate the suggestion to compare HFM with **MERU** and **HYPE**. Since **HYPE** only provides official checkpoints for the **ViT-L/14** backbone, we conduct all experiments using this architecture. Specifically, we full-parameter fine-tuned MERU and HYPE under 16-shot setting.  The results in Table R1 on 5 difficult datasets show that HFM significantly outperforms these representation-learning baselines (+23.0% and +14.1%), proving its effectiveness. We will add the comparison results in the revised manuscript.
>
> **Table R1. Comparison with existing hyperbolic VLM models.**
> |Model|Aircraft|SAT|DTD|SUN|UCF|Avg|
> |---|---|---|---|---|---|---|
> |MERU|34.7|89.6|59.6|54.9|61.7|60.1|
> |HYPE|42.0|91.5|65.3|69.2|76.9|69.0|
> |HFM|68.7|93.4|80.1|81.6|91.5|83.1|
>
> ### **Q3. Computational overhead.**
> Thanks for the insightful comment. We add a quantitative efficiency comparison between HFM and the Euclidean FMA baseline under a capacity-matched setting (same flow network, 512-dim features, batch size 32, and 10 inference steps on the same hardware). Overall, results in Table R2 show that HFM introduces only marginal parameter and memory overhead, while incurring a moderate runtime increase due to the additional hyperbolic geometric operations (e.g., log/exp maps and geodesic distance computation). We will add the computational overhead evaluation in the revised manuscript.
>
> **Table R2. Computational overhead comparison.**
> |Method|Params(M)|ParamMem(MB)|Train/step(ms)|PeakTrainMem(MB)|Infer(ms)|PeakInferMem(MB)|
> |---|---|---|---|---|---|---|
> |FMA|17.22|65.68|9.288|392.70|0.622|197.30|
> |HFM|17.28|65.93|11.774|409.87|0.739|208.92|
>
> ### **Q4. Limitations.**
> We agree that broader task generalization is important. However, extension of flow matching to other tasks is non-trivial as it requires re-modeling the flow dynamics to handle different supervision structures.
> - Image-Text Retrieval. Extending HFM to retrieval is challenging because current ODE formulation is inherently **1-to-1**, whereas retrieval is typically **1-to-many** (e.g., 5 captions per query image on Flickr30k). Training a 1-to-1 flow on such data causes the model to transport an image feature toward a drift caption target during inference. *This drift prevents the query image feature from maintaining a consistent alignment with all five valid caption targets simultaneously.* We consider the development of 1-to-many hyperbolic flow formulations a valuable direction for future exploration.
> - Dense Prediction. We extend HFM on few-shot semantic segmentation (PASCAL-5i) on Table R3, transitioning from global embeddings to **dense patch-level tokens**. In this setting, the flow transports initial query patch representations ($x_0$) toward support-conditioned target states ($x_1$). Specifically, we generate foreground/background prototypes via masked pooling on support tokens and fuse them with text. The query patches are projected onto the Lorentz manifold and transported along geodesic-consistent directions toward these targets. The refined dense representations are then decoded into the final mask.  As seen in Table R2, HFM consistently outperforms the CLIP-Lora baseline across all folds. This validates that hyperbolic flow matching effectively rectifies pixel-level trajectories even in high-dimensional dense spatial tasks.
>
> **Table R3.  Evaluation on dense prediction.**
> |Method|mIoU/FB-IoU(Fold0)|mIoU/FB-IoU(Fold1)|mIoU/FB-IoU(Fold2)|mIoU/FB-IoU(Fold3)|mIoU/FB-IoU(Avg)|
> |---|---|---|---|---|---|
> |CLIP-Lora|62.5/78.3|75.9/85.5|55.6/73.0|52.6/70.0|61.7/76.7|
> |HFM|63.7/78.8|76.9/85.9|60.1/74.1|52.7/70.6|63.4/77.4|

---

> > ### Author Rebuttal · Reviewer_mvKY · 2026-04-05
> >
> > We thank the authors for their effort in providing additional experiments and clarifications during the rebuttal period.
> > Q1 (Partially resolved). The authors provided additional explanations regarding the motivation and design of the proposed hyperbolic flow formulation. These clarifications help better contextualize the method, although the conceptual distinction from prior work combining flow-based alignment and hyperbolic representations could still be further discussed in the final version.
> > Q2 (Resolved). The additional comparison with prior hyperbolic VLM models (MERU and HYPE) is appreciated and helps better position the proposed approach with respect to related work.
> > Q3 (Resolved). The computational overhead analysis is helpful. The efficiency comparison between HFM and the Euclidean baseline provides useful quantitative evidence regarding runtime and memory usage.
> > Q4 (Partially resolved). The extension experiment on few-shot semantic segmentation (PASCAL-5i) is encouraging and suggests that the framework may generalize beyond classification. Further validation on additional tasks would strengthen this claim.
> > For these reasons, I respectfully maintain my score of 4.

---

> > > ### Author Response · Authors · 2026-04-06
> > >
> > > We sincerely appreciate your follow-up comments. We would like to solve the remaining concerns.
> > > ## **Q1. Conceptual novelty.**
> > > Compared with flow-based alignment, we would like to clarify our novelty from both **empirical evidence** and  **theoretical analysis**:
> > > - **Empirical Evidence.** HFM’s novelty is validated by:
> > >     - *Path Entanglement Issue*: We are the first to identify “Path Entanglement” in VLM adaptation. As shown by the **Trajectory Entanglement Rate** in **Table R5** (cf **Reviewer EaGV Q3**) and **Fig 3**, Euclidean FM suffers from severe trajectory collisions due to flat geometry.
> > >     - *Centripetal Geometric Necessity*: To prove the necessity of manifold, we apply the same objectives (Contrastive loss and Step-wise consistency) to Euclidean FMA*. In **Table R3** (cf **Reviewer bTtr Q2**), these yield negligible gains (+0.2%) compared to HFM’s significant lead (+2.1%), confirming that hyperbolic exponential volume is indispensable for decoupling.
> > >     - *Algorithmic Design (HFM vs. Velocity-based HFM)*: We show that simply FM in hyperbolic space with standard velocity supervision is insufficient. In **Table R4** (cf **Reviewer bTtr Q3**), a velocity-based HFM gets sub-optimal results. In contrast, our $\mathcal{L}_{step}$ provides +1.3% gain. This is because velocity errors in tangent spaces amplify exponentially due to negative curvature $\kappa$, causing trajectory drift. Our PO prevents this trajectory drift.
> > >     - *Adaptive Diameter-based Stopping*. Unlike FM's fixed-step (early) stop, our DS leverages our centripetal hierarchy to use semantic diameter as a dynamic threshold. It prevents features from drifting into incorrect clusters, getting +0.6% gain on **Table 2** across difficult datasets.
> > > - **Theoretical Analysis.** We prove the advantages of HFM over Euclidean FM by:
> > >     - *Volume Capacity*: In Euclidean space $\mathbb{R}^d$, the volume of a ball $V_E(r)$ with radius $r$ grows polynomially: $V_E(r) = \mathcal{O}(r^d)$. This causes inevitable path overlap at high class densities. Conversely, the Lorentz manifold $\mathcal{L}^{n,\kappa}$ exhibits **exponential volume growth**: $V_H(r) = \mathcal{O}(e^r)$. Anchoring textual roots at the origin and visual leaves at the boundary via CHA, HFM offers vast geometric room for trajectory decoupling
> > >     - *Geodesic Deviation*: Per the *Jacobi Equation*, the separation $J(t)$ between two nearby geodesics with curvature $-\kappa$ grows exponentially: $\frac{d^2 J}{dt^2} - \kappa J = 0 \implies J(t) \approx e^{(d-1)\sqrt{\kappa}t}$. It ensures that even small initial semantic differences are amplified during transport, allowing our PO to effectively confine flows within isolated geodesic corridors.
> > >     - *Formal Separation Gain*: The superiority over baseline is quantified by the ratio of geodesic separation $S(r)$ at distance $r$ for a given initial angular separation $\Delta \theta$: In Euclidean space: $S_E(r) = r \cdot \Delta \theta$. In the manifold: $S_H(r) = \frac{1}{\sqrt{\kappa}} \sinh(\sqrt{\kappa}r) \cdot \Delta \theta$. The gain is expressed as: $\text{Gain}(r) = \frac{S_H(r)}{S_E(r)} = \frac{\sinh(\sqrt{\kappa}r)}{\sqrt{\kappa}r} \approx \frac{e^{\sqrt{\kappa}r}}{2\sqrt{\kappa}r}$. This exponential gain $\mathcal{O}(e^{\sqrt{\kappa}r} / r)$ offers a superior geodesic margin, mathematically explaining the reduced Trajectory Entanglement Rate in *Table R2 in Reviewer EaGV Q3*.
> > >
> > > Compared with **static** hyperbolic representation learning (e.g., MERU, HYPE), HFM shifts the paradigm from “where features are” to “how features evolve”. It addresses the gap in **dynamic transport paths** during few-shot adaptation. Through our PO and DS designs, HFM makes features follow geodesic-consistent trajectories rather than relying on one-step static mappings, as validated by Table R1. Furthermore, HFM demonstrates **plug-and-play** extensibility: in **Table R4**, incorporating HFM into MERU/HYPE yields +2.5% and +1.7% gains, proving the universality across different hyperbolic backbones.
> > >
> > > **Table R4. Hyperbolic VLM models + HFM.**
> > > |Model|Aircraft|SAT|DTD|SUN|UCF|Avg|
> > > |---|---|---|---|---|---|---|
> > > |MERU|34.7|89.6|59.6|54.9|61.7|60.1|
> > > |+HFM|35.9|91.7|62.3|58.2|64.8|62.6|
> > > |HYPE|42.0|91.5|65.3|69.2|76.9|69.0|
> > > |+HFM|44.9|92.4|67.7|70.9|77.6|70.7|
> > >
> > > ## **Q4. Limitations.**
> > > We construct a 1-to-1 image-text retrieval setting on Flickr30k by keeping 1 caption per image and evaluating over 5 caption folds. In Table R5, HFM's consistent lead over CLIP-LoRA in both I2T and T2I proves its potential in instance-level tasks. We consider complex 1-to-many retrieval a key future work direction and will add **Limitation** section to discuss this exploration.
> > >
> > > **Table R5. 1-to-1 image-text retrieval (R@1).**
> > > |Method|I2T|T2I|
> > > |---|---|---|
> > > |CLIP-Lora|77.7|76.0|
> > > |HFM|78.2|77.1|
> > >
> > > We will add these discussions into revision. We hope our further clarification can fully address your concerns. If you have any other concerns, please do not hesitate to let us know.

---

### Official Review · Reviewer_5CsR · 2026-04-03

**Soundness:** 3
**Presentation:** 3
**Significance:** 3
**Originality:** 2
**Overall Recommendation:** 3
**Confidence:** 2

**Summary:**

This paper studies the few-shot adaptation problem and argues that the existing Euclidean-based FM suffers from path entanglement. Therefore, the paper proposes a path-decoupled hyperbolic flow matching framework, which employs feature transport on the Lorentz manifold. Meanwhile, the paper conducted experiments on 11 few-shot benchmarks, and the results seem that the proposed framework achieves better performance compared with baselines.

**Compliance With Llm Reviewing Policy:**

Affirmed.

**Final Justification:**

The paper proposed a path-decoupled hyperbolic FM framework for the few-shot adaptation problem, which is well motivated, and the idea is interesting. The paper is conducted on several datasets and demonstrates the effectiveness of the proposed framework. However,  both the theoretical and the parameter analysis are limited.

**Key Questions For Authors:**

1. Could the author provide more analysis or case studies for path entanglement? It is suggested to add these to better demonstrate the motivation.
2.  How sensitive is the method to key hyperparameters? i.e., curvature $\kappa$, scaling factors $\alpha_{txt}$ and the balance weight $\lambda$
3. What is the computational cost of HFM compared with baselines?
4. Could the author give a case or theoretical analysis to discuss whether HFM benefits from the hyperbolic space?

**Limitations:**

The impact statement is too brief and does not discuss possible limitations and impacts.

**Strengths And Weaknesses:**

Strengths
1. The paper is interesting and well motivated. It gives the limitation of prior Euclidean FM methods and introduces a hyperbolic framework on the Lorentz manifold.
2. The paper introduces three components for the motivation, and the paper is well written and easy to follow.
3. The paper conducts experiments on 11 benchmarks, and the proposed framework achieves better avg. performance compared with baselines.

Weaknesses
1. The theoretical analysis remains limited. The paper gives the empirical results of the proposed framework. It is suggested to give a theoretical analysis to demonstrate the improvement.
2. The novelty is related to prior FM methods, especially FMA. It is suggested to give more analysis on what is new compared with existing FM methods in hyperbolic space.
3. Some hyper-parameters are not sufficiently analysed, i.e., $\phi(N)=0.5\log_10(N)$, $\alpha_{txt}=0.5\alpha_{img}$. How to choose these hyperparameters. It is suggested to add more hyper-parameter analysis about the framework.
4. The paper studies a hyperbolic framework with three components, and the computational cost is not discussed.

---

### Decision · Program_Chairs · 2026-04-30

**Decision:**

Accept (regular)

**Comment:**

All three on-time reviewers agree that the paper is technically sound, clearly presented, and demonstrates consistent empirical improvements across a broad suite of few-shot benchmarks. The rebuttal successfully addressed the main concerns raised during the review process, and Reviewer bTtr updated their recommendation from Weak Accept to Accept. Overall, the paper presents a coherent framework that is both practically deployable and empirically validated, addressing an important problem in few-shot VLM adaptation.
The ACs recommend acceptance.
(Reviewer 5CsR is disregarded due to late submission.)